



**CALIBRATING ELECTROMAGNETIC INDUCTION CONDUCTIVITIES WITH TIME-DOMAIN**
**REFLECTOMETRY MEASUREMENTS**
Dragonetti[1] Giovanna, Alessandro Comegna[2], Ali Ajeel[2], Gian Piero Deidda[3], Nicola
Lamaddalena[1], Giuseppe Rodriguez[4], Giulio Vignoli[3,5], Antonio Coppola[2]*
(1) Mediterranean Agronomic Institute (MAIB) - Land & Water Department, Valenzano (Bari),
Italy
(2) University of Basilicata, School of Agricultural, Forestry and Environmental Sciences -
Hydraulics and Hydrology Division, Potenza, Italy. e-mail: antonio.coppola@unibas.it
(3) Dipartimento di Ingegneria Civile, Ambientale e Architettura, Università di Cagliari, Cagliari,
Italy
(4) Dipartimento di Matematica e Informatica, Università di Cagliari, Cagliari, Italy
(5) Groundwater and Quaternary Geology Mapping Department, Geological Survey of Denmark
and Greenland, Aarhus, Denmark
**Abstract**
This paper deals with the issue of monitoring the horizontal and vertical distribution of bulk
electrical conductivity, $\sigma_b$, in the soil root zone by using Electromagnetic Induction (EMI)
sensors under different water and salinity conditions. In order to deduce the actual distribution
of depth-specific $\sigma_b$ from EMI depth-weighted apparent electrical conductivity (EC$_a$)
measurements, we inverted the signal by using a regularized 1D inversion procedure designed
to manage nonlinear multiple EMI-depth responses. The inversion technique is based on the
coupling of the damped Gauss-Newton method with truncated generalized singular value
decomposition (TGSVD). The ill-posedness of the EMI data inversion is addressed by using a
sharp stabilizer term in the objective function. This specific stabilizer promotes the
reconstruction of blocky targets, thereby contributing to enhance the spatial resolution of the
EMI reconstruction. Time-Domain Reflectometry (TDR) data are used as ground-truth data for
calibration of the inversion results. An experimental field was divided into four transects 30 m
long and 2.8 m wide, cultivated with green bean and irrigated with water at two different
salinity levels and using two different irrigation volumes, to induce different salinity and water
contents within the soil profile. For each transect, 26 regularly spaced monitoring sites (1 m
apart) were selected for soil measurements using a Geonics EM-38 and a Tektronix
Reflectometer. Despite the original discrepancies in the EMI and TDR data, we found a
significantly high correlation of the means and standard deviations of the two data series,
especially after filtering the TDR data. Based on these findings, the paper introduces a novel
methodology to calibrate EMI-based electrical conductivity via TDR direct measurements by
simply using the statistics of the two data series.

**Introduction**
Soil water content and salinity vary in space both vertically and horizontally. Their distribution
depends on management practices and on the complex nonlinear processes of soil water flow
and solute transport, resulting in variable storages of solutes and water (Coppola et al. 2015).





Monitoring the actual distribution of water and salts in the soil profile explored by roots is crucial to managing irrigation with saline water, while still maintaining an acceptable crop yield. For monitoring water and salts over large areas, there are now non-invasive techniques based on electromagnetic sensors which allow the bulk electrical conductivity of soils, $\sigma_b$, to be determined (Sheets and Hendrickx 1995, Corwin and Lesch 2005, Robinson et al. 2012, Doolittle and Brevik 2014, von Hebel et al. 2014) among many others). $\sigma_b$ depends on soil water content, $\theta$, electrical conductivity of the soil solution (salinity), $\sigma_w$, tortuosity of the soil-pore system, $\tau$, and other factors related to the solid phase such as bulk density, clay content and mineralogy.

Electromagnetic induction (EMI) sensors provide measurements of depth-weighted apparent electrical conductivity, $EC_a$, according to the specific depth distribution of the soil bulk electrical conductivity, $\sigma_b$, as well as the depth response function of the sensor used (McNeill 1980). Thus dependence on $\sigma_b$ makes $EC_a$ sensitive to soil salinity and water content. In principle, specific procedures for estimating salinity and water content may be developed through controlled laboratory experiments where $\sigma_b$, $\sigma_w$ and $\theta$ are measured simultaneously (Rhoades and Corwin 1981). That said, to monitor salinity and water content, it is crucial to correctly infer the depth-distribution of $\sigma_b$ from profile-integrated $EC_a$ readings.

To date, this issue has been tackled by applying two different strategies: The first is to use empirical calibration relations relating the depth-integrated $EC_a$ readings to the $\sigma_b$ values measured by alternative methods - like Time-Domain Reflectometry (TDR) -within discrete depth intervals (Rhoades and Corwin 1981, Lesch et al. 1992, Triantafilis, Laslett, and McBratney 2000, Amezketa 2006, Yao and Yang 2010, Coppola et al. 2016); The second consists in the 1D inversion of the observations from the EMI sensor to reconstruct the vertical conductivity profile (Borchers, Uram, and Hendrickx 1997, Hendrickx et al. 2002, Santos et al. 2010, Lavoué et al. 2010, Mester et al. 2011, Minsley et al. 2012, Deidda, Fenu, and Rodriguez 2014, von Hebel et al. 2014).

With regard to $EC_a$ inversion, a forward model still commonly used is the cumulative response model or local-sensitivity model (LSM) (McNeill 1980). McNeill's linear approach is well suited to the cases characterized by an induction number, $\beta$ (defined as the ratio between the coil distance and the skin depth), much smaller than 1. However, because of increasing computing power, improved forward modeling algorithms based on more accurate nonlinear approaches are becoming increasingly common (Hendrickx et al. 2002, Deidda, Fenu, and Rodriguez 2014, Deidda, Bonomi, and Manzi 2003, Lavoué et al. 2010, Santos et al. 2010). For example, these more sophisticated forward modeling codes can cope with a wider range of conductivities for which the assumption $\beta \ll 1$ is not necessarily met.

To obtain reliable vertical distributions of electrical conductivity, the $EC_a$ data used for the inversion should consist of multi-configuration data. Hence, data collection should be performed either with the simultaneous use of different sensors or with different acquisition configurations with only one sensor (different configurations may consist, e.g., in different coil orientations, varying intercoil separations and/or frequencies – see, for example Díaz de Alba and Rodriguez (2016)). Multi-configuration data can be effectively used to invert for vertical




electrical conductivity profiling since the $EC_a$ measures actually investigate different,
overlapping soil volumes. Devices specifically designed for the simultaneous acquisition of
multi-configuration data are currently available. Some of them consist of one transmitter and
several receivers with different coil separations and orientations (Santos et al. 2010). If,
instead, a sensor with a single intercoil distance is available, a valid alternative to having multi-
configuration measurements could be to record the data at different heights above the ground.
Unfortunately, like every other physical measurement, frequency-domain electromagnetic
measurements are sensitive to noise that is very hard to model effectively. Therefore, for
example, as discussed in Lavoué et al. (2010), Mester et al. (2011), and von Hebel et al. (2014),
an instrumental shift in conductivity values could be observed due to system miscalibration and
the influence of surrounding conditions such as temperature, solar radiation, power supply
conditions, the presence of the operator, zero-leveling procedures, cables close to the system
and/or the field setup (see, amongst others, (Sudduth, Drummond, and Kitchen 2001, Robinson
et al. 2004, Abdu, Robinson, and Jones 2007, Gebbers et al. 2009, Nüsch et al. 2010). Therefore
the $EC_a$ data from EMI measurements would generally need proper calibration. One option
could be to use soil cores as ground-truth data. In this case, $EC_a$ measurements at the sampling
locations are compared against $EC_a$ data predicted by the theoretical forward response applied
to the true electrical conductivity distribution measured directly on the soil cores (Triantafilis,
Laslett, and McBratney 2000, Moghadas et al. 2012). Clearly, this strategy is extremely time-
(and resource-) consuming. To avoid drilling, Lavoué et al. (2010) introduced a calibration
method, later also adopted by Mester et al. (2011) and von Hebel et al. (2014), using the
electrical conductivity distribution obtained from Electrical Resistivity Tomography (ERT) data
as input for electromagnetic forward modeling. The $EC_a$ values predicted on the basis of ERT
data were used to remove the observed instrumental shift and correct the measured
conductivity values by linear regression. However, a prerequisite for such an approach
concerns the reliability of the inversion of the ERT result. This is not only due to the quality of
the original data, but also the adopted inversion procedure. Indeed, ERT inversion is an ill-
posed problem: its solutions are characterized by non-uniqueness and instability with respect
to the input data (Yu and Dougherty 2000, Zhdanov 2002, Günther 2011). In the Tikhonov
regularization framework, ill-posedness is addressed by including the available prior
information. Such information can be very general. For example, it can be geometrical (i.e.,
associated to the presence of smooth or sharp boundaries between different lithologies).
Clearly, the final result largely reflects the initial guess formalized via the chosen regularization
term (Pagliara and Vignoli 2006, Günther 2011, Vignoli, Deiana, and Cassiani 2012, Fiandaca et
al. 2015).
When relatively shallow depths have to be explored (1-2m), direct soil sampling and ERT can be
effectively replaced by TDR observations. In this line of reasoning, this paper focuses on the use
of TDR data to calibrate $EC_a$ measurements obtained via EMI. To do this, a dataset collected
during an experiment carried out along four transects under different salinity and water
content conditions (and monitored by both EMI and TDR sensors) will be utilized. We first
tackle the problem of inferring the soil electrical conductivity distribution from multi-height $EC_a$



readings via the proper inversion strategy. Then we assess the quality of these reconstructions
by using TDR data as ground-truth. In this respect, in the following, we discuss how to
effectively compare the $\sigma_b$ values generated by the EMI inversion with the associated TDR
values. In fact, as discussed by (Coppola et al. 2016), because of their relatively smaller
observation volume, TDR data provide quasi-pointlike measurements and do not integrate the
small-scale variability (of soil water content, solute concentrations, etc.) induced by natural soil
heterogeneity. By contrast, EMI data necessarily overrule the small-scale heterogeneities seen
by TDR probes as they investigate a much larger volume. Accordingly, the paper provides a
methodology to calibrate EMI results by TDR readings. This procedure lies in conditioning the
original TDR data and in the statistical characteristics of the two EMI and TDR data series. On
the basis of the proposed analysis we discuss the physical reasons for the differences between
EMI and TDR-based bulk electrical conductivity and identify a method to effectively transfer the
reliable TDR information across the larger volume investigated by EMI.
**Materials and Methods**
The experiment was carried out at the Mediterranean Agronomic Institute of Bari (MAIB) in
south-eastern Italy. The soil was pedologically classified as Colluvic Regosol, consisting of a
silty-loam layer of an average depth of 0.6 m on fractured calcarenite bedrock. The
experimental set-up (Figure 1) consisted of four transects of 30 m length and 2.8 m width,
equipped with a drip irrigation system with five dripper lines at 0.35 m distance and a distance
among drippers along each line of 0.2 m, with a dripper discharge of 2 l/h. Green beans were
grown in each transect. The irrigation volumes were calculated according to the time-dynamics
of water content in the first 0.25 m measured by a TDR probe inserted vertically at the soil
surface. TDR readings were taken: i) just before and ii) two hours after every irrigation. Based
on the difference between the water content at field capacity and that measured just before
irrigation the volumes to bring the soil water content back to the field capacity were able to be
calculated.
The four transects were irrigated with water at two different salinity levels and with two
different water volumes. Transect 1: 100% of the irrigation water at 1 dSm$^{-1}$ (hereafter 100-
1dS); Transect 2: 50% of irrigation water at 1 dSm$^{-1}$ (50-1dS); Transect 3: 100% of the irrigation
water at 6 dSm$^{-1}$ (100-6dS); Transect 4: 50% of irrigation water at 6 dSm$^{-1}$ (50-6dS). Water
salinity was induced by adding calcium chloride (CaCl$_2$) to tap water. Irrigation volumes were
applied every two days.
EMI readings in both horizontal (EC$_a$H) and vertical magnetic dipoles (EC$_a$V) configurations were
collected by using a Geonics EM38 device (Geonics Limited, Ontario, Canada). The EM38
operates at a frequency of 14.6 kHz with a coil spacing of 1 m, and with an effective
measurement depth of ≈0.75 m and ≈1.5 m, respectively, in the horizontal and vertical dipole
configurations (McNeill, 1980). The lateral footprint of the EM38 measurement can be
considered approximately equal to the vertical one. Thus, the $\sigma_b$ seen by the EMI in a given
discrete depth-layer differs from that seen by a TDR probe in the same depth-layer, due to the
very different spatial resolutions.





At the beginning of each measurement campaign, the sensor was "nulled" according to the
manufacturer's manual. Readings were taken just after each irrigation application at 1 m step,
along the central line of each transect, for an overall total of 26 measurements per transect,
per campaign. Taking measurements just after irrigation allowed relatively time-stable water
contents to be assumed at each site throughout the monitoring campaigns.
Multi-height EM38 readings were performed at 26 locations in the middle line of each transect
during the growing season. Readings were acquired at heights of 0.0, 0.2, 0.4 and 0.6 m from
the ground. Overall, seven EM38 measurement campaigns were carried out during the
experiment, from July 7[th] to September 2[nd].
Just after each EM38 measurement campaign, a TDR probe was inserted vertically at the soil
surface (0.0-0.25 m) in 26 sites, each corresponding to the central point of an EM38 reading. A
Tektronix 1502C cable tester (Tektronix Inc., Baverton, OR) was used in this study. It enables
simultaneous measurement of water content, $\theta$, and bulk electrical conductivity, $\sigma_b$, of the soil
volume explored by the probe (Heimovaara et al. 1995, Robinson and Friedman 2003, Coppola
et al. 2011, Coppola et al. 2015). The TDR transmission line consisted of an antenna cable
(RG58, 50 $\Omega$ characteristic impedance, 2 m long and with 0.2 $\Omega$ connector impedance) and
three-wire probes, 0.25 m long, 0.07 m internal distance, and 0.005 m in diameter. The TDR
probe was not embedded permanently at fixed depths along the soil profile to avoid any
potential disturbance to the EMI acquisitions.
Only immediately after the last EM38 campaign (September 2[nd]) were TDR readings taken at
three different depth intervals (0.0-0.2, 0.2-0.4, 0.4-0.6 m). After the measurements at the
surface (0.0-0.2 m), a trench was dug up to 0.2 m depth. TDR probes were then inserted
vertically for the additional collection of the data in the interval 0.2-0.4 m, after which the
trench was deepened up to 0.4 m and readings were taken at 0.4-0.6 m. $\sigma_{b,TDR}$ readings in this
last campaign were used for the calibration of the EM38 data. All the remaining six data series
will be used for a validation study of the approach developed in this paper (which will be the
subject of a follow-up paper).
Data Handling
*Multi-height EMI readings inversion*
Nonlinear 1D forward modeling, which predicts multi-height EMI readings from a loop-loop
device, can be obtained by suitable simplification of Maxwell's equations that takes the
symmetry of the problem into account. This approach is described in detail in (Hendrickx et al.
2002), and is based on a classical approach extensively described in the literature (Wait 1982,
Ward and Hohmann 1988). The predicted data are functions of the electrical conductivity and
the magnetic permeability in a homogeneously and horizontally layered medium.
When the coils of the recording device are vertically oriented with respect to the ground
surface, the reading at height $h$ can be expressed by using the integral:

$$-\rho^3 \int_0^\infty \lambda^2 e^{-2h\lambda} R_0(\lambda) J_0(\rho\lambda) d\lambda, \tag{1}$$





where $\rho$ denotes the distance between the coils, $J_0(\lambda)$ is the Bessel function of the first kind
of order 0, and $R_0(\lambda)$ is a complex valued function which depends upon the electromagnetic
properties of the ground layers. A similar expression is valid also when the coils are horizontally
aligned. Hence the dependence of the measured data on the electrical conductivity $\sigma_k$, of the
(homogeneous) j-th layer is incorporated into the function $R_0(\lambda)$. We discretize the problem
with n layers whose characteristic parameters $\sigma_j$ (with j = 1, . . ., n) are the unknowns we invert
for. In the present research, we neglect any dependence of the electromagnetic response on
magnetic permeability as we assume it is fixed and equal to the permeability of empty space.
We consider two measurements for each location: one for the horizontal and one for the
vertical configuration of the transmitting and receiving loops. In this way, the data used as
inputs for the inversion are 2m, where m is the number of heights $h_1$, $h_2$, . . ., $h_m$ where the
measurements are performed.
A least squares data fitting approach leads to the minimization of the function:

$$f(\sigma) = \frac{1}{2} \sum_{i=1}^{2m} r_i^2(\sigma), \tag{2}$$

where $\sigma = (\sigma_1, \ . \ . \ ., \ \sigma_n)^T$, and $r_i^2(\sigma)$ is the misfit between the *i-th* measurement and the
corresponding forward modeling prediction based on equation (1).
We solve the nonlinear minimization problem by the inversion procedure described in Deidda,
Fenu, and Rodriguez (2014). The algorithm is based on a damped regularized Gauss-Newton
method. The problem is linearized at each iteration by means of a first order Taylor expansion.
The use of the exact Jacobian (whose expression is detailed in Deidda, Fenu, and Rodriguez
(2014) makes the computation faster and more accurate than using a finite difference
approximation. The damping parameter is determined in order to ensure both the convergence
of the method and the positivity of the solution. The regularized solution to each linear
subproblem is computed by the truncated generalized singular value decomposition (TGSVD -
Díaz de Alba and Rodriguez (2016) employing different regularization operators. Besides the
classical regularization matrices based on the discretization of the first and second derivatives,
to further improve the spatial resolution of EMI inversion results, we tested a nonlinear
regularization stabilizer promoting the reconstruction of blocky features (Zhdanov, Vignoli, and
Ueda 2006, Ley-Cooper et al. 2015, Vignoli et al. 2015, Vignoli et al. 2017). The advantage of
this relatively new regularization is that, when appropriate prior knowledge about the medium
to reconstruct is available, it can mitigate the smearing and over-smoothing effects of the more
standard inversion strategies. This, in turn, can make the calibration of the EMI data against the
TDR data more effective. For this reason, in the following, the EMI results used for our
assessments are those inferred by means of this sharp regularization. The differences between
the "standard" smooth (based on the first derivative) reconstruction and the sharp one are
clearly shown in Figure 2.



It is worth noting that the constant magnetic permeability assumption is not always valid:
inverting for the magnetic permeability is sometimes not only necessary, but can also provide
an additional tool for soil characterization (Deidda, Diaz De Alba, and Rodriguez 2017).
For the sake of clarity, hereafter, the $\sigma_b$ values generated from the EMI data inversion will be
identified explicitly as $\sigma_{b,EMI}$.
*TDR-based water content and bulk electrical conductivity*
The Tektronix 1502C can measure the total resistance, $R_t$, of the transmission line by:

$$R_t = Z_c \frac{(1+\rho_\infty)}{(1-\rho_\infty)} = R_s + R_c \tag{3}$$

where $R_s$ is the soil's contribution to total resistance and $R_c$ accounts for the contribution of the
series resistance from the cable, the connector, $Z_c$, is the characteristic impedance of the
transmission line, and $\rho$ is a reflection coefficient at a very long time, when the waveform has
stabilized.
The $\sigma_b$ value at 25°C can be calculated as (Rhoades and van Schilfgaarde 1976, Wraith et al.
252 1993):

$$\sigma_{b\,25°C} = \frac{K_c}{Z_c} f_T \tag{4}$$

where $K_c$ is the geometric constant of the TDR probe and $f_T$ is a temperature correction factor
to be used for values recorded at temperatures other than 25°C. Both $Z_c$ and $K_c$ can be
determined by measuring $R_t$ with the TDR probe immersed in a solution with known
conductivity $\sigma_b$. Hereafter, these $\sigma_b$ measurements will be identified as $\sigma_{b,TDR}$.
*Evaluation of Concordance between $\sigma_{b,TDR}$ measurements and $\sigma_{b,EMI}$ estimates*
The agreement between $\sigma_{b,TDR}$ measurements and $\sigma_{b,EMI}$ estimations in the 0.0-0.20 m layer was
evaluated by the Concordance Correlation Coefficient, $\rho_L$:

$$\rho_L = \frac{2s_{xy}}{z_x^2 + z_y^2 + (m_x - m_y)^2} \tag{5}$$

where $m_x$, $m_y$, $s_x$, $s_y$, $s_{xy}$ are means, standard deviations and covariances of the two data series
($x = \sigma_{b,EMI}$; $y = \sigma_{b,TDR}$), respectively.
Scatter plots of the $\sigma_{b,EMI}$ and $\sigma_{b,TDR}$ data series (both original and filtered) for the depth interval
0.0-0.20 m were evaluated by the line of perfect concordance (1:1 line) and the reduced major
axis of the data (RMA) (Freedman et al. 1991). The method combines measurements of both
precision and accuracy to determine how close the two data series are to the line of perfect
concordance $\sigma_{b,EMI} = \sigma_{b,TDR}$. Compared to the classical Pearson correlation coefficient, $\rho_P$:

$$\rho_P = \frac{s_{xy}}{s_x s_y}, \tag{6}$$

$\rho_L$ not only measures the strength of linear relationship (how close the data in the scatter plot
are to a line), but also the level of agreement (how close that line is to the line of perfect
agreement, the 1:1 line). In this sense, $\rho_L$ may also be calculated as (Cox 2006):





$$\rho_L = \rho_P C_b,$$

$$C_b = \frac{2}{(v + 1/v + u^2)},$$

$$v = s_x / s_y,$$

(7)

and

$$u = (m_x - m_y)/\sqrt{s_x s_y},$$

where $C_b$ is the bias correction factor measuring how far the best-fit line deviates from the 1:1 line. The maximum value of $C_b = 1$ ($0 < C_b < 1$) corresponds to no deviation from the line. The smaller $C_b$ is, the greater the deviation from the line. In other words, $C_b$ is a measure of accuracy (how much the average estimate differs from the average measurement value, assumed to be the true value) and refers to the systematic error, whereas $\rho_P$ is a measure of precision (measures the variability of measurements around their own average) and refers to the random error. The RMA line is given by:

$$y = (m_y - \beta m_x) + \beta x = \alpha + \beta x.$$

(8)

This line passes through the means of the x and y values and has slope given by the sign of Pearson's correlation coefficient, $\rho_P$, and the ratio of the standard deviations, s, of the two series (Freedman et al. 1991, Corwin and Lesch 2005):

$$\beta = s_y / s_x.$$

(9)

$\rho_L$ increases in value as (i) the RMA approaches the line of perfect concordance (a matter of accuracy) and (ii) the data approach the RMA (a matter of precision). In the ideal case of perfect concordance, the intercept of the RMA, α, should be 0 and $\beta$ should be 1. Therefore, α ≠ 0 or $\beta$ ≠ 1 indicate additive and/or multiplicative biases (location and/or scale shifts). The concordance was evaluated for the original TDR data, as well as for the filtered TDR data. For the analysis carried out in the results section, it is worth noting here that the coefficients α and $\beta$ depend only on the statistical characteristics (mean and standard deviation) of the two series, as $\alpha = m_y - \beta m_x$ and $\beta = s_y / s_x$.

*Fourier filtering*

Because of their relatively small observation volume ($\approx 10^{-3}$ m$^3$), TDR sensors provide quasi-pointlike measurements and are thus more effective in capturing small-scale variability (in water content, solute concentrations) induced by natural soil heterogeneity. Thus the variability within a set of TDR readings is expected to originate from a combination of small and large-scale heterogeneities (high and low spatial frequency components). By contrast, the EMI measurements (because of the size and physics of the instrumentation) necessarily integrate out the small-scale variability at the TDR scale of investigation.

Hence, in order to make the two datasets comparable, the original spatial TDR data series need to be filtered to remove the variation from small-scale heterogeneities (recorded only by the TDR probe). In this way, only the information at a spatial scale equal to or larger than the observation volume of both sensors is preserved.



The Fourier transform (FT) of discrete stationary series of length M equispaced at intervals Δp
($x_p$, p=0,1,…,M-1) (where x is the variable, and p the spatial or temporal location on the series)
is defined as (Shumway 1988):

$$X(k) = M^{-1} \sum_{s=0}^{M-1} (x_p - \bar{x}) \exp(-2\pi i v_k p), \tag{10}$$

where k= 0,1….,M-1, X(k) are the Fourier coefficients, i= $\sqrt{-1}$ , $v_k$ = k/M is the wave number (or
frequency) in cycles per unit distance (or time) and $\bar{x}$ is the sample mean. If the series is
detrended, $x_p$ in equation (22) is the detrended series.
The FT in equation (10) may be written in terms of sine and cosine transform, noting that:

$$\exp(-2\pi i v_k p) = \cos(-2\pi v_k p) - i\sin(-2\pi v_k p) \tag{11}$$

Thus equation (10) becomes:

$$X(k) = X_C(k) - iX_S(k) \tag{12}$$

The Fourier coefficients X(k) are complex numbers. Most software packages (e.g., MatLab, SAS,
Microsoft Excel) have built-in Fast Fourier transform (FFT) algorithms that considerably speed
up the computation of equation (10); the sine and cosine transforms are immediately available
from the real and imaginary parts of the computed X(k).
By using the following coefficients:

$$a_k = -\frac{2}{M} \text{imag}(X(k)), \quad 0 < k < \frac{M}{2};$$
$$b_k = -\frac{2}{M} \text{real}(X(k)), \quad 0 < k < \frac{M}{2}, \tag{13}$$

it is easy to perform the inverse FT and recover the original signal:

$$x(p) = a_0 + \sum_{k=0}^{(M-1)/2} (a_k \sin(2\pi v_k p) + b_k \cos(2\pi v_k p)) \tag{14}$$

Equation (14) is central to the filtering approach we use in the present paper. It can be used to
reconstitute a smoothed data series by retaining selected harmonics alone (e.g., only the low
frequency harmonics). The frequencies to be selected can be identified by examining the
power spectral density - see equation (16) below - of the data series.
The periodogram can be written as the squared modulus of the FT:

$$P_x(v_k) = |X(k)|^2 = [X_C^2(k) + X_S^2(k)] = X(k)\overline{X(k)}, \tag{15}$$

where the overbar denotes complex conjugate. $P_x$ is an asymptotically unbiased estimator for
the spectrum (Shumway 1988). It is common practice to average adjacent values of the
periodogram to obtain estimates with more degrees of freedom, and create a smoothed power
spectrum. The average spectral estimator, in a frequency interval centered on $v_k$, is defined as:

$$f_x^{P,B}(v_k) = L^{-1} \sum_{l=-(L-1)/2}^{(L-1)/2} P\left(v_k + \frac{l}{M}\right) = L^{-1} \sum_{l=-(L-1)/2}^{(L-1)/2} |X(k+l)|^2, \tag{16}$$

where L is some odd integer considerably smaller than M and defining the size of the averaging
window. Hence, the averaging window is characterized by a bandwidth B = L/M (cycles per



point) centered on $v_k$. $f_x^{P,B}(v_k)$ is the periodogram-based power spectrum averaged on B and
with, approximately, a chi-squared distribution, in which the degrees of freedom depend on
the width L of the window used.
The 100(1-α) confidence interval for the smoothed spectrum can be calculated as:

$$\frac{2Lf_x^{P,B}(v_k)}{\chi_{2L}^2(a/2)} \leq f_x^n(v_k) \leq \frac{2Lf_x^{P,B}(v_k)}{\chi_{2L}^2(1-a/2)}, \tag{17}$$

where α is the significance level and $f_x^n(v_k)$ is the background noise power spectrum. The null
hypothesis is $f_x^{P,B}(v_k) = f_x^n(v_k)$ vs. $f_x^{P,B}(v_k) \neq f_x^n(v_k)$. If $f_x^n(v_k)$ falls within the interval in equation
(17), we fail to reject the hypothesis. If not, the estimated power spectrum at a given frequency
$v_k$ has to be considered significantly different from that of the assumed background noise. In
the case of white noise, implying a uniform distribution of the power spectrum across
frequencies, $f_x^n(v_k)$ can be considered as the mean of all power spectrum estimates.

**Results and Discussion**
Hereafter, the original and filtered data will be respectively labeled ORG and FLT. The graphs in
panel (a) of Figures 3, 4 and 5 compare $\sigma_{b,TDR}$ measured by TDR against the corresponding
conductivity $\sigma_{b,EMI}$ retrieved by the EMI (sharp) inversion, respectively, for the layers at 0-0.20,
0.20-0.40, and 0.40-0.60 m. From the left, the graphs refer respectively to the transects
identified as 100-6dS, 50-6dS, 100-1dS and 50-1dS. All the plots report the line of perfect
concordance (1:1, black line) and the main regression axis (MRA, red line).
The general outcome is that, in all four transects, and for all three considered depth-layers, the
$\sigma_{b,EMI}$ values underestimate the $\sigma_{b,TDR}$ measurements, such that the MRA line generally lies
above the 1:1 line. Not surprisingly, the EMI result seems quite insensitive to TDR variability.
Also, a considerable scatter around the MRA line may be observed for all four transects.
Tables 1, 2 and 3 show the MRA coefficients ($C_b$, α , β), as well as the Pearson, $\rho_P$, and the
concordance correlation, $\rho_L$, for the three depth-layers and for all four transects investigated.
We recall that the bias correction factor, $C_b$, the slope, β, and the intercept, α, should be
respectively close to 1, 1 and 0, for the MRA to approximate the line of perfect concordance.
For all the transects and all the depth-layers considered, the parameters confirm the relatively
loose relationship between $\sigma_{b,EMI}$ and $\sigma_{b,TDR}$ already observed in the graphs, both in terms of
accuracy (the distance of the MRA line from the 1:1) and precision (the data scatter around the
MRA line).
von Hebel et al. (2014) found a similar behavior when comparing their EMI and ERT datasets. In
that case, the $EC_a$ values measured by EMI systematically underestimated the $EC_a$ generated by
applying EMI forward modeling to the $\sigma_b$ distribution retrieved by ERT. To remove the bias, the
authors simply performed a linear regression between measured and predicted ECa after
applying a ten-term moving average to the original data. By using the regression coefficients,
all the measured $EC_a$ values were converted to ERT-calibrated $EC_a$ values.
Here, we follow a different approach to calibrate the $\sigma_{b,EMI}$ values against the $\sigma_{b,TDR}$
measurements based on the MRA coefficients and hence on the statistical parameters (mean
and standard deviation) of the two data series. Specifically, the present approach looks for a
systematic correction of the bias based on well-defined statistical sources of the discrepancies.
In short, the proposed method performs the calibration in the $\sigma_b$ model-space, instead of the
$EC_a$ data-space.
Our model-space approach mostly relies on the statistical parameters of the two series.
Analyzing the role of these statistics in explaining the discrepancies between EMI and TDR data
observed in Figures 3-5 may help to understand how they can be exploited for converting EMI
measurements to TDR values.
In nearly all of the graphs in panel (a) of Figures 3-5, the discrepancies between $\sigma_{b,EMI}$ and $\sigma_{b,TDR}$
values can be decomposed in the following components:
1. The distance along the $\sigma_{b,EMI}$ axis of the MRA line from the 1:1 line, that is the difference
between the $\sigma_{b,EMI}$ and the $\sigma_{b,TDR}$ means.
2. The difference in the slope of the MRA and of the 1:1 lines, which stems from the different
variability of $\sigma_{b,EMI}$ (its standard deviation) and that of $\sigma_{b,TDR}$. We recall here that the slope of
the MRA is just the ratio of the two standard deviations, $\hat{\beta} = s_y / s_x$.
3. The scatter of the data around the MRA line, which may come from different sensors' noise
and the influence of surrounding conditions (e.g., temperature).
Below, we analyze in detail the role of all these three points with the support of the measured
data.
1. The distance of the MRA from the 1:1 line may be mostly ascribed to the difference in the
observed means. The graph in Figure 6a compares the means for the two original series (open
squares-solid line for TDR, open circles-dashed line for EMI). The plot in Figure 6b reports the
same comparison on a 1:1 plot (open triangles-solid regression line). The means confirm the
general underestimation of TDR by the EMI data. However, the trends are evidently similar,
which is reflected in the high correlation between the means of the two series, with a
significantly high $R^2$=0.81. The high correlation of the means has very positive implications from
an applicative point of view, as, after calibration in a specific soil, it allows the TDR mean to be
inferred given the mean of EMI readings taken in that soil, and thus gives us the possibility to
transpose the more reliable TDR information across the larger area that can be practically
investigated with EMI.
2. The different slope of the two lines has to be ascribed to the different variability of the two
series. The graph in Figure 7a compares the standard deviations for the two original series
(open squares-solid line for TDR, open circles-dashed line for EMI). The graph in figure 7b
reports the same comparison on a 1:1 plot (open triangles-solid regression line). Conceptually,
the different variability of the two series may well be related to the different sensor
observation volumes (coming from the different spatial sensitivity of the sensors) (Coppola et
al. 2016). For TDR probes, most of the measurement sensitivity is close to the rods (Ferre et al.
1998). Conversely, the spatial resolution of inverted EMI ECa values may be much lower as the



resolution of the EMI result depends on the physics of the method, the specifications (and
configuration) of the recording device, and the regularization type applied during inversion.
That said, the EMI is generally unable to capture the small-scale variability seen by the TDR. For
our calibration purposes, it is important to make the variability of EMI and TDR conductivities
actually comparable. As discussed by Coppola et al. (2016), a method can be to filter the high
frequency component (at small spatial scale) of the original TDR data, while retaining the lower
frequency information, that is information at a spatial scale larger than the observation volume
of the TDR sensor and attuned with the resolution of the EMI distribution values coming from
the inversion. From a practical point of view, this makes sense, as TDR readings are often "too
local" to actually represent the macroscopic physical characteristics of interest for applications
(water content, solute concentrations). The volume explored by a TDR probe may, or may not,
include preferential channels (Mallants et al. 1994, Oberdörster et al. 2010), stones (Coppola et
al. 2011, Coppola et al. 2013), small-scale changes in the texture and structure (Coppola et al.
2011), which can make the interpretation of local measurements difficult for practical
applications. In this sense, EMI's removal of these small-scale effects may be desirable from a
management perspective.
Accordingly, original TDR data were conditioned via Fourier filtering, as described in the
Material and Methods section. The number of low-frequency harmonics to be used for
rebuilding the filtered signal was selected according to the spectrum for each depth and
transect - see equation (16) - and, in general, it was included in three-six harmonics. The
filtering results, in terms of standard deviations, are reported in figure 7a (crosses-dashed line)
and in figure 7b (open squares-dashed regression line). As expected, filtering made the
standard deviations much closer (almost overlapping in many cases) for all transects and for all
considered depth-layers. The regression improved significantly from 0.25 for the original data
to 0.78 when TDR data were filtered. As with the means, the high correlation of the standard
deviations has positive implications from a practical point of view: it allows the TDR standard
deviation to be inferred, given the standard deviation of EMI readings taken in that soil. Panel b
of Figures 3 to 5 shows the comparison of the original EMI and filtered TDR data. The
concordance coefficients in the case of filtered TDR data are again reported in Tables 1 to 3.
Obviously, because of the almost overlapping EMI and TDR standard deviations after filtering,
the MRA line turned out to be much more parallel to the 1:1 line, as indicated by the
coefficient $\beta$, which is now much closer to 1.
3. In general, however, filtering left the scatter around the MRA line almost unaltered. Here the
scatter was zeroed by again using the intercept and the slope coefficients of the MRA obtained
after TDR filtering. Specifically, the filtered TDR data were recalculated from the original EMI
data as:

$$\sigma_{b,TDR(FLT)}^{rg} = \alpha + \beta\sigma_{b,EMI} \tag{18}$$

The superscript *rg* means filtered data after regression. The results are again reported in panel
c of figures 3-5. As an example of the calibration results, figure 8 compares the maps of bulk
electrical conductivity for the 100-6dS transect obtained respectively by plotting the original





$\sigma_{b,EMI}$ (figure 8a) coming from the inversion of the EMI signal and the calibrated $\sigma_{b,TDR(FLT)}^{rg}$
(figure 8b) obtained by applying the equation 18 to the $\sigma_{b,EMI}$ data of the first map. After
calibration, the nearly homogeneous $\sigma_b$ distribution represented in the map of figure 8a,
coming from the substantial insensitivity of the original EMI data to TDR variability, turn into a
physically more plausible $\sigma_b$ layering, largely reproducing the true one observed by the TDR
probes .
All the points discussed above provide the rationale to deduce the TDR-FLT data based on the
statistical parameters of the EMI and TDR data ($m_x$, $m_y$, $s_x$, $s_y$). Summarizing, the procedure
requires the following steps:
1. Filtering the TDR data, by retaining only the low-frequency part of the signal. The number of
harmonics to be selected depends on the length of the data series, as well as on the spectrum
characteristics. This step will make the standard deviations of the two data series similar, thus
turning the data parallel to the 1:1 line;
2. Using the average ($m_x$, $m_y$) and the standard deviation ($s_x$, $s_y$) of the original $\sigma_{b,EMI}$ EMI data
and of the filtered $\sigma_{b,TDR(FLT)}$ TDR data to calculate the MRA line coefficients as $\alpha = m_y - \beta m_x$ and
$\beta = s_y / s_x$. Of course, the averages for the original and the filtered TDR data will coincide;
3. Straightening the data on the MRA line (zeroing the scatter) by recalculating the TDR-FLT
data from the original EMI data and the MRA coefficients $\sigma_{b,TDR(FLT)}^{rg} = \alpha + \beta \sigma_{b,EMI}$.
As already discussed, the high correlation of the means and the standard deviations of the two
series are central for this procedure to be of practical interest. To explain this with an example,
let us assume an experiment (like that described herein) has been carried out in a calibration
field within the area to be monitored by an EMI sensor; the experiment would allow
regressions to be built for the mean and the standard deviation of the original EMI and the
filtered TDR, like those shown in figures 6b and 7b.
Now let us take a set of ECa readings in the area to be monitored. After inversion, these ECa
data provide a set of $\sigma_{b,EMI}$ values. For the reasons discussed above, we know that these values
do not represent the actual values one would measure directly by using a TDR probe. Rather,
they only contain the low-frequency information supplied by TDR (most likely, together with
some shifts connected with the poor absolute calibration of the EMI system and/or the working
conditions, e.g., the temperature). We now have a workflow to convert these $\sigma_{b,EMI}$ data into
the corresponding filtered TDR values. In other words, the proposed workflow enables us to
translate the original non-calibrated $\sigma_{b,EMI}$ data into the actual $\sigma_b$ we would collect in ideal
conditions, and which would perfectly match "low-resolution" TRD measurements. The
workflow requires:
1. The mean and the standard deviation of EMI, which can be calculated by the $\sigma_{b,EMI}$ data;
2. The mean and the standard deviation of filtered TDR, which can be calculated by the
regressions from the calibration experiment (as in figures 6b and 7b);
These statistics may now be used to evaluate coefficients α and β to be used in equation (18) to
convert the original $\sigma_{b,EMI}$ data into as many $\sigma_{b,TDR(FLT)}^{rg}$ values. Hence, $\sigma_{b,TDR(FLT)}^{rg}$ is our best





possible estimation of the true electrical conductivity at the scale of investigation of the EMI
survey: it is the original $\sigma_{b,EMI}$ after the application of the appropriate rescaling and shifts
deduced by the more reliable and absolutely calibrated TDR measurements.
**Conclusions**
The objective of the paper was to infer the bulk electrical conductivity distribution in the root
zone from multi-height (potentially non-calibrated) EMI readings. TDR direct measurements
were used as ground-truth $\sigma_b$ data to evaluate the correctness of the $\sigma_b$ estimations generated
by EMI inversion. For all four transects and for all three depth-layers considered in this study,
the $\sigma_{b,EMI}$ values underestimate the $\sigma_{b,TDR}$ measurements, such that the MRA line generally lies
above the 1:1 line. Also, a considerable scatter around the MRA line was observed for all four
transects.
The proposed analysis allowed discussion of the physical reasons for the differences between
EMI- and TDR-based electrical conductivity and develop an approach to calibrate the original
$\sigma_{b,EMI}$ by using the $\sigma_{b,TDR}$ measurements. Our approach is based on the MRA coefficients and
hence on the statistical parameters (mean and standard deviation) of the two series.
Specifically, the approach looks for a systematic correction of the bias based on well-defined
statistical sources of the discrepancies. A significant high correlation was found for the means
and the standard deviations of the two series, especially after filtering the TDR data. This is
crucial for the practical application of our methodology.
The proposed strategy lies in the fact that TDR direct measurements supply absolutely
calibrated observations of the electrical conductivity of the soil and hence can be effectively
used to calibrate the conductivity distributions inferred from EMI data. The availability of EMI
calibrated data paves the way to reliable reconstructions of the electrical conductivity
distribution over large areas (typical for EMI surveys, but not for TDR campaigns) unaffected by
the usual EMI miscalibrations. This, in turn, can result in the possibility of effective time-lapse
surveys and/or in consistent merging of subsequent surveys (at any time the dynamic
components of the system under investigation can be neglected).
On the other hand, the proposed statistical workflow for making the TDR measurement
comparable with the associated EMI results provides a more sophisticated approach than
simple smoothing to upscale the TDR data. Thus, from the opposite perspective, the approach
in question can be used to tackle the problems connected with handling the TDR data
characterized by excessively high spatial resolution.
Finally, the approach used here allows TDR calibration measurements to be used not
necessarily at the same sites and in the same quantities as EMI readings, as it is based on
means and standard deviations and does not require site-by-site data comparison.





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



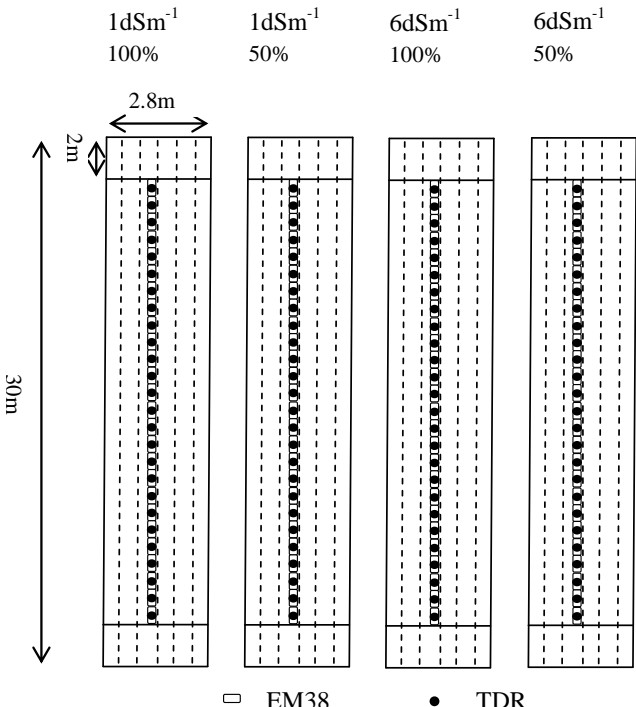


Figure 1. Schematic view of the experimental field




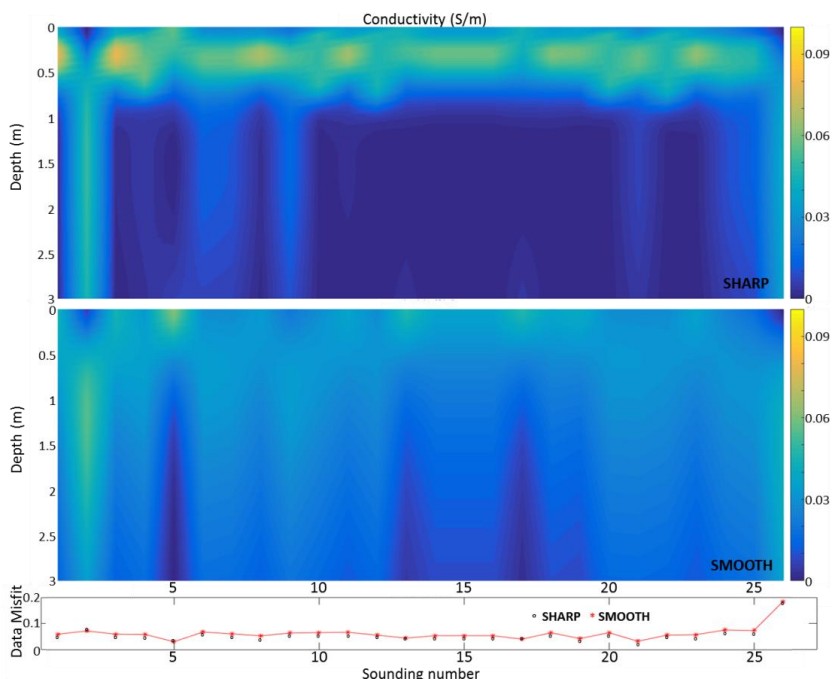


Figure 2. Examples of sharp and smooth inversions applied to the same dataset 100-6dS. The
results are shown together with their corresponding data misfit









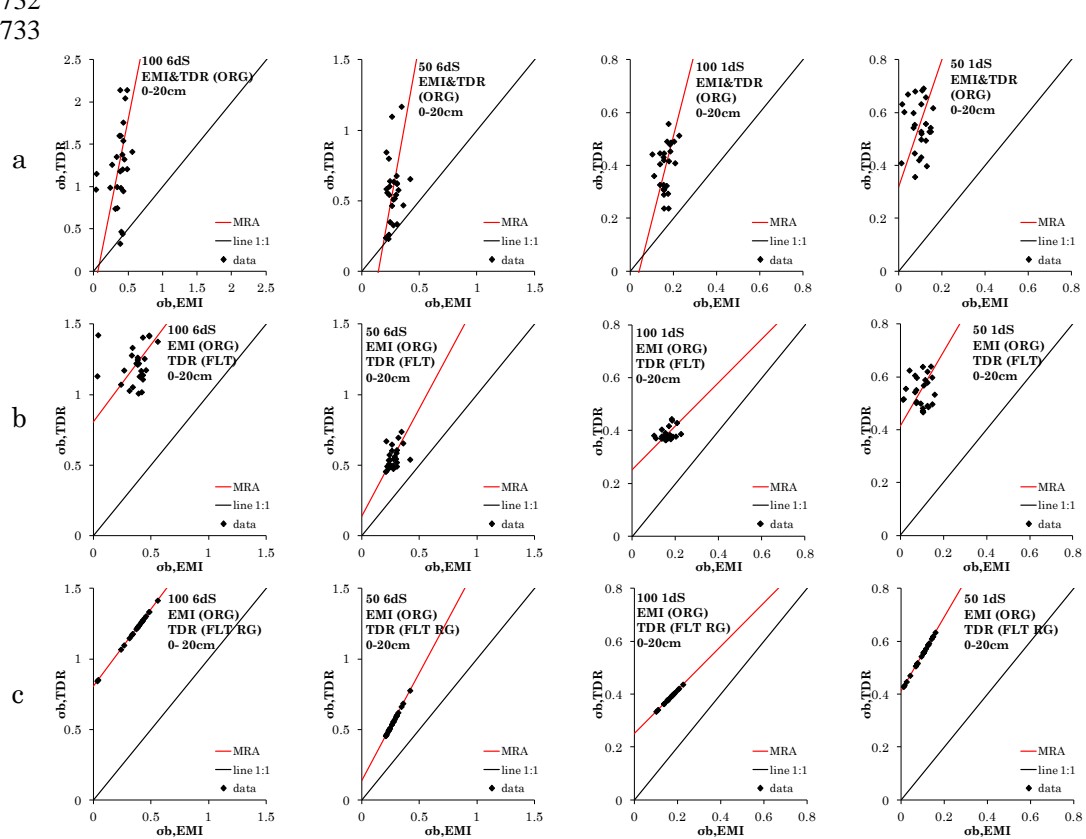


Figure 3. Comparison between $\sigma_{b,TDR}$ and $\sigma_{b,EMI}$ for all four transects for the depth layer 0-20 cm.
The graphs in the horizontal panels are respectively for: (a) Original EMI and TDR data; (b)
original EMI and filtered TDR data (c) original EMI and filtered TDR data after regression (RG)
based on MRA parameters





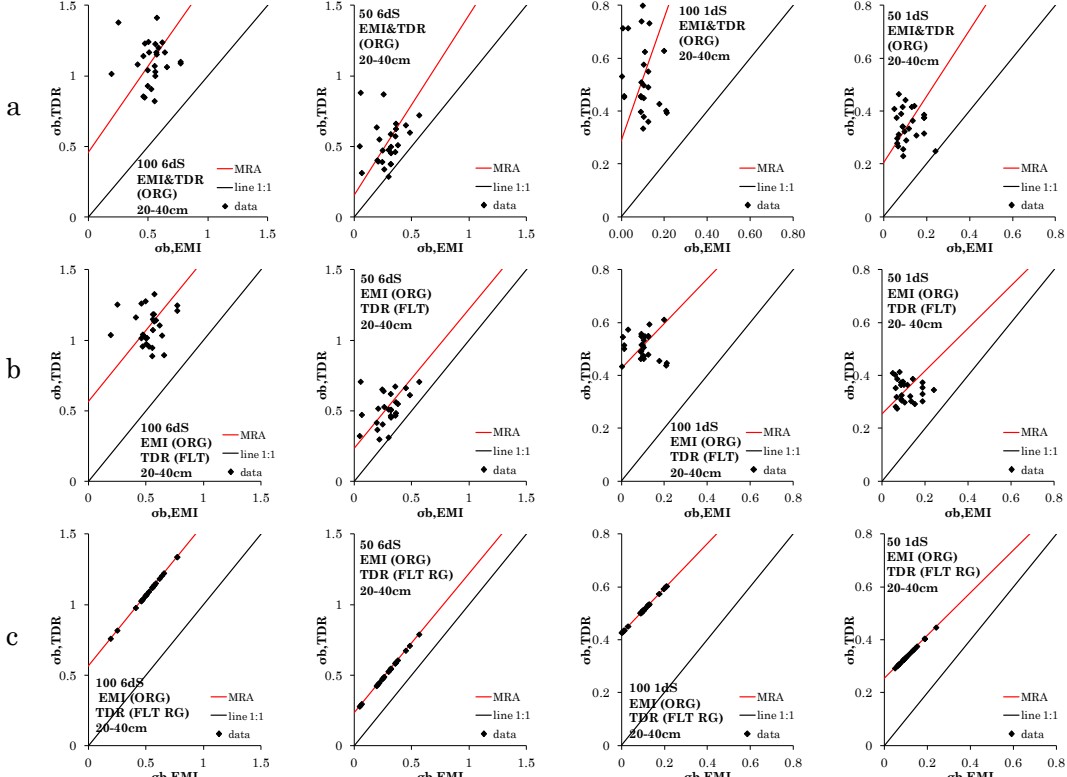

Figure 4. Comparison between $\sigma_{b,TDR}$ and $\sigma_{b,EMI}$ for all four transects for the depth layer 20-40 cm. The graphs in the horizontal panels are respectively for: (a) Original EMI and TDR data; (b) original EMI and filtered TDR data (c) original EMI and filtered TDR data after regression (RG) based on MRA parameters





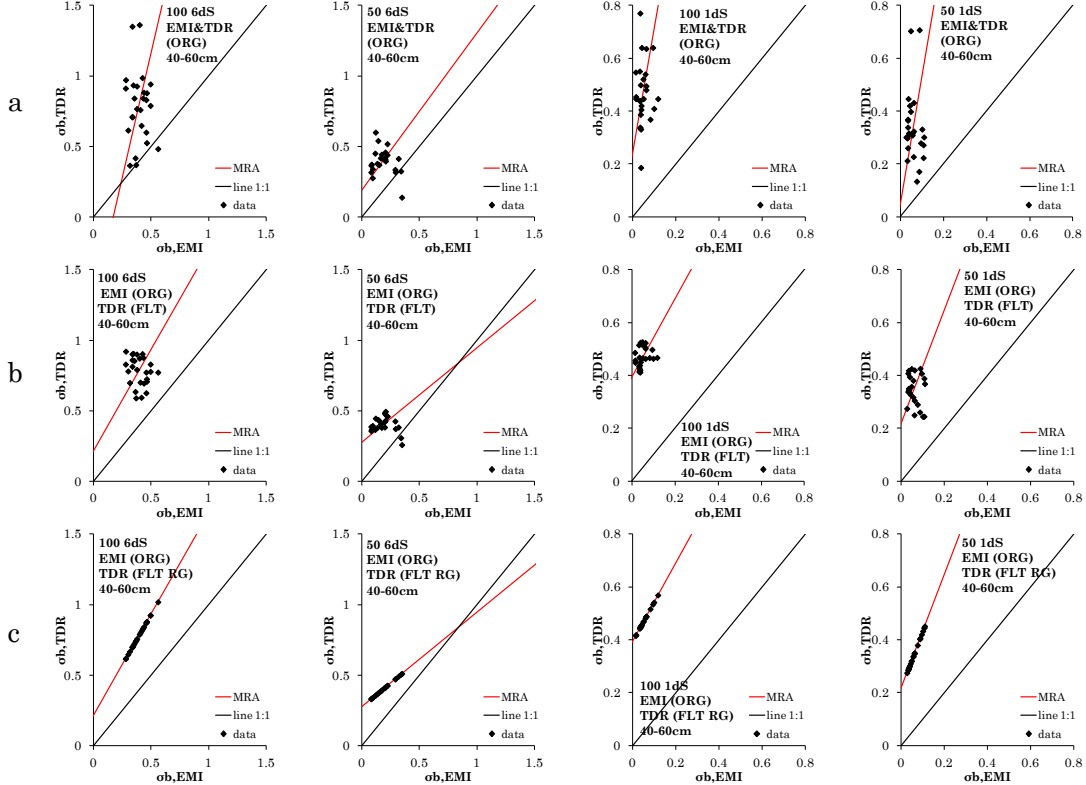

Figure 5. Comparison between $\sigma_{b,TDR}$ and $\sigma_{b,EMI}$ for all four transects for the depth layer 40-60 cm. The graphs in the horizontal panels are respectively for: (a) Original EMI and TDR data; (b) original EMI and filtered TDR data (c) original EMI and filtered TDR data after regression (RG) based on MRA parameters





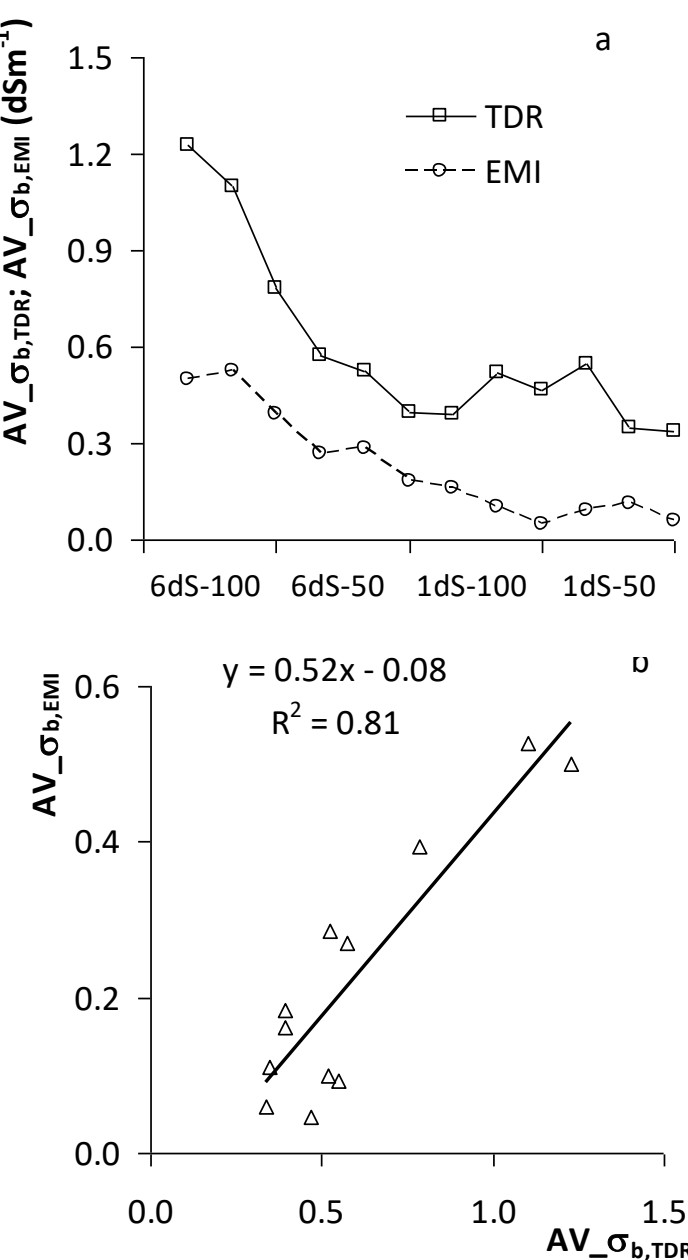

Figure 6. (a) Comparison of the means for the two original series (open squares-solid line for TDR, open circles-dashed line for EMI); (b) The same comparison on a 1:1 plot (open triangles-solid regression line). In figure 6a the four treatments are shown in sequence. For each treatment, the three values are for the three depths (0-20, 20-40 and 40-60 cm)




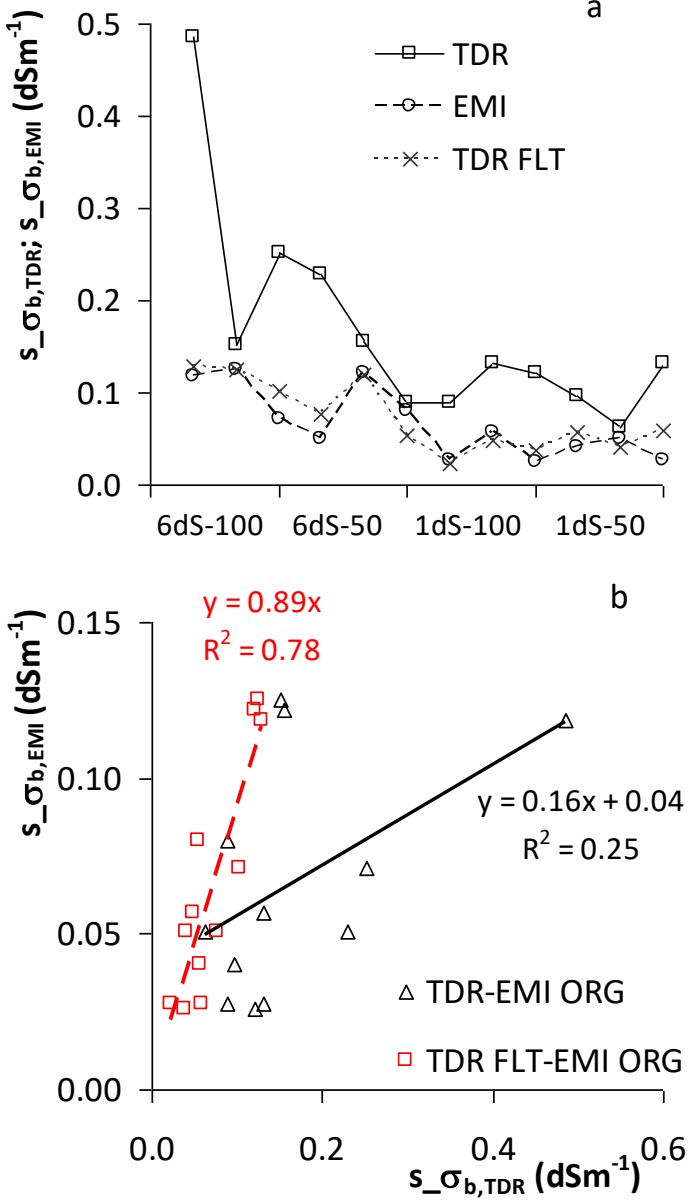

Figure 7. (a) Comparison of the standard deviations of the TDR original series (open squares-solid line), of the EMI original series (open circles-dashed line) and of the filtered TDR series (crosses-dashed line); (b) The same comparison on a 1:1 plot: original TDR and EMI data (open triangles-solid regression line); filtered TDR and original EMI data (open squares-dashed regression line). In figure 7a the four treatments are shown in sequence. For each treatment, the three values are for the three depths (0-20, 20-40 and 40-60 cm)





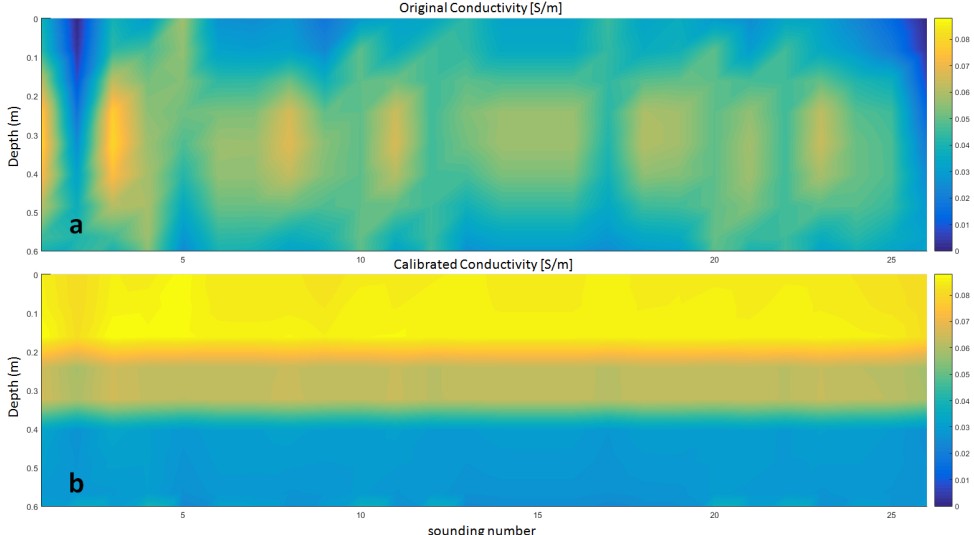

Figure 8. Maps of bulk electrical conductivity for the 100-6dS transect obtained respectively by plotting the original $\sigma_{b,EMI}$ (a) coming from the inversion of the EMI signal and the calibrated $\sigma_{b,TDR(FLT)}^{rg}$ (b) obtained by applying the equation 18 to the $\sigma_{b,EMI}$ data of the first map





Table 1. Concordance parameters for the four transects at depth 0-20 cm. The table reports the Concordance, $\rho_L$, and the Pearson, $\rho_P$, correlation, as well as parameters α and β of the MRA line. The bias factor, $C_b$, is also shown.

| Graph panel | $C_b$ 20cm | $\rho_L$ 20cm | $\rho_P$ 20cm | β 20cm | α 20cm |
|---|---|---|---|---|---|
| **1dS-100** | | | | | |
| a | 0.08 | 0.02 | 0.31 | 3.20 | -0.13 |
| b | 0.02 | 0.01 | 0.35 | 0.82 | 0.25 |
| c | 0.02 | 0.02 | 0.96 | 0.82 | 0.25 |
| **1dS-50** | | | | | |
| a | 0.04 | 0.0002 | -0.01 | 2.39 | 0.32 |
| b | 0.02 | 0.0006 | 0.03 | 1.40 | 0.41 |
| c | 0.02 | 0.02 | 0.96 | 1.4 | 0.41 |
| **6dS-100** | | | | | |
| a | 0.12 | 0.03 | 0.25 | 4.10 | -0.27 |
| b | 0.04 | 0.005 | 0.12 | 1.09 | 0.81 |
| c | 0.04 | 0.04 | 0.96 | 1.09 | 0.81 |
| **6dS-50** | | | | | |
| a | 0.16 | 0.03 | 0.22 | 4.52 | -0.65 |
| b | 0.09 | 0.04 | 0.42 | 1.52 | 0.14 |
| c | 0.09 | 0.08 | 0.96 | 1.52 | 0.14 |



Table 2. Concordance parameters for the four transects at depth 20-40 cm. The table reports the Concordance, $\rho_L$, and the Pearson, $\rho_P$, correlation, as well as parameters α and β of the MRA line. The bias factor, $C_b$, is also shown.

| Graph panel | $C_b$ 40cm | $\rho_L$ 40cm | $\rho_P$ 40cm | β 40cm | α 40cm |
|---|---|---|---|---|---|
| **1dS-100** | | | | | |
| a | 0.08 | -0.02 | -0.21 | 2.32 | 0.29 |
| b | 0.03 | -0.002 | -0.07 | 0.84 | 0.43 |
| c | 0.03 | 0.03 | 0.96 | 0.84 | 0.43 |
| **1dS-50** | | | | | |
| a | 0.10 | -0.004 | -0.04 | 1.25 | 0.21 |
| b | 0.07 | -0.01 | -0.13 | 0.81 | 0.25 |
| c | 0.07 | 0.07 | 0.96 | 0.81 | 0.25 |
| **6dS-100** | | | | | |
| a | 0.10 | 0.001 | 0.01 | 1.21 | 0.46 |
| b | 0.09 | 0.004 | 0.05 | 0.99 | 0.57 |
| c | 0.09 | 0.08 | 0.96 | 0.99 | 0.57 |
| **6dS-50** | | | | | |
| a | 0.40 | 0.06 | 0.15 | 1.27 | 0.16 |
| b | 0.35 | 0.14 | 0.39 | 0.98 | 0.24 |
| c | 0.35 | 0.34 | 0.96 | 0.98 | 0.24 |





Table 3. Concordance parameters for the four transects at depth 40-60cm. The table reports the Concordance, $\rho_L$, and the Pearson, $\rho_P$, correlation, as well as parameters α and β of the MRA line. The bias factor, $C_b$, is also shown.

| Graph panel | $C_b$ 60cm | $\rho_L$ 60cm | $\rho_P$ 60cm | β 60cm | α 60cm |
|---|---|---|---|---|---|
| **1dS-100** | | | | | |
| a | 0.03 | 0.002 | 0.07 | 4.69 | 0.25 |
| b | 0.01 | 0.003 | 0.24 | 1.48 | 0.40 |
| c | 0.01 | 0.01 | 0.96 | 1.48 | 0.40 |
| **1dS-50** | | | | | |
| a | 0.08 | -0.01 | -0.12 | 4.81 | 0.05 |
| b | 0.04 | -0.01 | -0.17 | 2.14 | 0.22 |
| c | 0.04 | 0.04 | 0.96 | 2.14 | 0.22 |
| **6dS-100** | | | | | |
| a | 0.16 | -0.01 | -0.09 | 3.52 | -0.60 |
| b | 0.09 | -0.02 | -0.25 | 1.43 | 0.22 |
| c | 0.09 | 0.08 | 0.96 | 1.43 | 0.22 |
| **6dS-50** | | | | | |
| a | 0.24 | -0.07 | -0.27 | 1.11 | 0.19 |
| b | 0.15 | -0.03 | -0.18 | 0.67 | 0.28 |
| c | 0.15 | 0.15 | 0.96 | 0.67 | 0.28 |