# Peer review of "CALIBRATING ELECTROMAGNETIC INDUCTION CONDUCTIVITIES WITH TIME-DOMAIN"

_Hydrology and Earth System Sciences, 2017_

## Referee Comment (RC1) · G. Cassiani (Referee) · 18 Jun 2017

I have found the paper interesting and potentially worth publication. However I find it somewhat surprising that the authors seem to believe that TDR is a better method than EMI to measure electrical conductivity. This seems to be an assumption made a priori, and not supported either by the scientific literature nor by any evidence in the paper. EMI is designed specifically to measure electrical conductivity, while TDR is designed with the measurement of dielectric properties in mind. Using TDR also to measure electrical conductivity can be done, similarly to using attenuation in GPR measurements to do the same. However it is not a recommended approach. My suggestion to

the author is to reverse the line of reasoning, believe more in EMI (with some caveats especially concerning the depth of investigation) and rather question TDR as a method for sigma measurement. In a nutshell, give more credibility to geophysics and question some belief in soil science. To this end, I also suggest that an eye is given to ERT as a technique that can provide ground truth much more reliable that TRD for electrical conductivity (see e.g. Cassiani et al., 2012 and Ursino et al., 2014, but many other papers deal with the EMI-ERT obvious relationship). I am also very surprised that moisture content estimates from TDR are not considered at all in the paper – yet the data must be available. I suggest the authors present also those (much more solid, I presume) data. I encourage the authors to revise the paper along these lines and resubmit this potentially interesting dataset. Line 26: "contributing to enhance the spatial resolution of the EMI reconstruction". I am not sure one can claim that the use of a stabilizer (how much needed would also require a specific discussion) truly enhances spatial resolution of a geophysical method. In my opinion this statement is wrong. I suggest a reformulation here. Line 35: "after filtering the TDR data." Even though this is the abstract, the statement is far too generic. Details about the filtering approach shall be briefly given here. Line 125: "Then we assess the quality of these reconstructions by using TDR data as ground-truth." This is a very brave statement. I do not see TDR as any more reliable to measure sigma than EMI, indeed quite the opposite. Line 132: "Accordingly, the paper provides a methodology to calibrate EMI results by TDR readings." This should not (cannot) be the focus of this paper. If the authors believe this is a viable strategy, I totally disagree. Line 291 and following. Spending time describing Fourier transformation is probably useless. Rather, I would concetrate on describing in detail what type of filtering is applied. "Fourier filtering" is unclear. I presume it is a spatial filtering made to enhance the long wavelengths? Please be more specific and try and link the approach to established (there are far too many) filtering techniques. Line 573: "Ferre" is actually "Ferré" Line 727 Figure 2. "Examples of sharp and smooth inversions applied to the same dataset 100-6dS. The results are shown together with their corresponding data misfit". I see only one curve of data misfit. Does it refer to

[Figure]

both sharp and smooth inversion? Also, I find it a bit difficult to justify in the images how some dark blue areas in the smooth inversion indeed correspond to slightly less dark blue areas in the sharp inversion. I am also a bit skeptical of the fact that using an EM38 one can image with confidence to a depth as large as 3 m! Line 735, Figure 3: here too some details about the filter applied to the TDR data shall be given. It is not acceptable that in a caption only the term "filtered" is applied. One can use any type of filter! The same applies to Figures 4 and 5

Figure 6: the difference between TDR and EMI measured sigma is quite large indeed. Overall I am not sure that TDR is the best method to measure sigma. Indeed it is not. TDR is the chief approach to measure dielectric properties.

Figure 8: the difference between the two images is striking. I am not sure how the authors are so confident that the correction applied to obtain the revised EMI image is correct. References Cassiani G., N. Ursino, R. Deiana, G. Vignoli, J. Boaga, M. Rossi, M. T. Perri, M. Blaschek, R. Duttmann, S. Meyer, R. Ludwig, A. Soddu, P. Dietrich and U. Werban, 2012, Non-invasive monitoring of soil static characteristics and dynamic states: a case study highlighting vegetation effects, Vadose Zone Journal, Special Issue on SPAC - Soil-plant interactions from local to landscape scale, August 2012, V.11, vzj2011.0195, doi: 10.2136/2011.0195. Ursino N., G. Cassiani, R. Deiana, G. Vignoli and J. Boaga, 2014, Measuring and Modelling water related soil – vegetation feedbacks in a fallow plot, Hydrology and Earth System Sciences (HESS), doi:10.5194/hess-18-1105-2014.

---

## Referee Comment (RC2) · Anonymous Referee #2 · 7 Jul 2017

TDR conductivity measurements show larger conductivity close to the surface as it should be expected since it is closer to the source of conductive material (added saline water). For this aspect, it is acceptable to consider TDR conductivity measurements as providing more reliable information, which could serve to calibrate "less reliable conductivities" obtained with EMI measurements (or their inversion). I therefore agree with the overall approach. I find the paper clear and well-written. I have nevertheless few questions at the EMI inversion stage which, I think, should be explored, or at least discussed before acceptance. I also think that Authors should show the final results for the other transects in order to see if the calibration performs well for the different irrigation experiments. This being said, this manuscript is interesting as it addresses

the problem of relating ground truth and EMI output with a pragmatic approach.

Non-uniqueness of the conductivity model resulting from the inversion:

One particular choice of the present study (compared to other cited studies) is to calibrate the conductivity after inversion instead of EMI apparent conductivity data (Eca). However, the argument of non-uniqueness of the inverse problem, which is actually used by the authors in the introduction to question a calibration with ERT method, could be used here in the same way to question the presented method.

This said, are you sure that the selected solution obtained using sharp regularization is the best solution to be compared with TDR? Actually, some of the smooth models (e.g. sounding numbers 5 , 13 17 in Figure 2) show better vertical concordance with what should be expected and with the best misfit. Anyway, Figure 2 clearly highlights the non-uniqueness of the problem, because it is possible to obtain very different models at similar misfits. Why, for example, not putting some effort to stabilize the regularization of the smooth inversion in order to have sounding N5-like results all along the transect? I do see that, because of fixed interfaces and a very little number of layers, your parameterization does not allow enough model space flexibility to have stable smooth results along the transect. But there are many ways to fix such issues (like, for example, among others, increasing the number of layers or applying some lateral constraints). I would also like to point out that the smooth misfits are slightly higher than for the sharp method (of 1-2 %, except for the few soundings mentioned above. This feature supports my previous comments), which is not in favor of the smooth regularization in a context of fair comparison. For all this, I would not qualify the smooth inversions presented here as a "standard" approach as it is not used with an optimal way (you use few layers with fixed interfaces). For Figure 2, I really recommend to plot the profiles of collected EMI data together with modeled (after inversion) data to see exactly how well all the channels were fitted. This would allow to properly evaluate and discuss the two methods of inversions. This is also important for a second aspect : in a cultivated area, I would expect some lateral anisotropy of the conductivity of the soil

layer due the preferential orientation of the lines of the agricultural work. If this is true, inverting HCP and VCP together would not be an optimal choice as such geometries induce eddy current with different preferential directions. Did you try to consider HCP only, and/or VCP only? Maybe there would be a better qualitative and quantitative consistency between TDR and EMI inversion results?

To sum up, I would have interpreted the results shown in Figure 2 in a different manner. After acceptation of the non-uniqueness of the EMI data inversion, I would have tried to find the right parameterization and regularization to get sounding N5-like results along the transect before starting the comparison with TDR. As a consequence, I believe that the calibration procedure presented in this study also corrects error due to the non-uniqueness inherent to the considered inverse problem (in addition to the spatial fractality problem already discussed in the text). And in the eventual case of lateral anisotropy, it maybe also correct for less realistic EMI results resulting from joint HCP and VCP inversion. However this two features means that the overall calibration procedure could be dependent on the initial method of inversion. In my opinion, this aspect should be explored in this study, or at least discussed in the text.

Application of the method to the four transects:

I think you should show the resulting sections (like Figure 8) for all the transects in order to check the consistency of the calibration procedure over larger areas (and implicitly for broader geological/irrigation settings) as it is claimed in the conclusion.

Some minor comments:

In the discussion about magnetic permeability (p7 line 239-241), you cite a non-published paper (Deidda et al, submitted). Here, it is necessary to also cite already published papers on this topic. There are a couple of recent studies dealing with the inversion of in-phase data for retrieving the magnetic permeability for the case of small EMI sensors.

Figure 8 shows spectacularly that the TDR conductivity of the first layer is largely underestimated by the EMI sharp inversion results. Are we sure that this first layer is well constrained by the considered EMI vertical soundings? Maybe it would be good to show and analyze the a priori covariance on model parameter associated to the selected 4-heights/2-geometries data set.

Summary of my recommendations:

Plot the measured data versus the modeled data in Figure 2. Explore or discuss the dependence of the overall procedure on the method of inversion. Show the final results for the four transects to confirm the robustness of the method on different irrigation contexts.

Sincerely

---

## Author Comment (AC1) · 15 Sep 2017

**Reply to Reviewer #1**

We thank Rev#1 for his valuable comments which allow us to improve our paper clarifying some issues, even though we partially disagree with him. We will describe our points of view in detail below. In the following, the original review is quoted in *italics*.
His main remarks concern:

1. The reliability of the TDR-based bulk electrical conductivity measurements
2. The explanation of the filtering procedure we used to make TDR and EMI measurements comparable.

1. The reliability of the TDR-based bulk electrical conductivity measurements

*I have found the paper interesting and potentially worth publication. However I find it somewhat surprising that the authors seem to believe that TDR is a better method than EMI to measure electrical conductivity. This seems to be an assumption made a priori, and not supported either by the scientific literature nor by any evidence in the paper. EMI is designed specifically to measure electrical conductivity, while TDR is designed with the measurement of dielectric properties in mind. Using TDR also to measure electrical conductivity can be done, similarly to using attenuation in GPR measurements to do the same. However it is not a recommended approach. My suggestion to the author is to reverse the line of reasoning, believe more in EMI (with some caveats especially concerning the depth of investigation) and rather question TDR as a method for sigma measurement. In a nutshell, give more credibility to geophysics and question some belief in soil science. To this end, I also suggest that an eye is given to ERT as a technique that can provide ground truth much more reliable that TDR for electrical conductivity (see e.g. Cassiani et al., 2012 and Ursino et al., 2014, but many other papers deal with the EMI-ERT obvious relationship).*

The reviewer states that "*EMI is designed specifically to measure electrical conductivity, while TDR is designed with the measurement of dielectric properties in mind*". This is not completely true. EMI actually measures the real (or the in-phase) and the imaginary (or the quadrature) parts of the ratio of the secondary to the primary magnetic field. Under certain constraints the imaginary part of this ratio, multiplied by an instrumental constant, gives the apparent electrical conductivity (McNeill, 1980), which, however, is not the electrical conductivity. In fact, EMI data must be inverted (and in some cases calibrated too) to get the electrical conductivity.
We agree with the reviewer that TDR is designed to measure the dielectric properties of soils. To be more specific, however, it should be said that TDR actually measures the apparent permittivity, which is defined as

$$\kappa_a = \frac{\mu_r \kappa'}{2}\left(1+\sqrt{1+\left(\frac{\kappa''}{\kappa'}\right)^2}\right)$$

where $\mu_r$ is the relative magnetic permeability while $\kappa'$ and $\kappa''$ are the real and the imaginary parts of the complex permittivity, respectively. The real part $\kappa'$ (also known as the dielectric constant) accounts for the energy stored in the dielectrics; the imaginary part $\kappa''$, which accounts for the energy dissipation, is defined as

$$\kappa'' = \varepsilon_{relax} + \frac{\sigma_{dc}}{\omega\varepsilon_0} = \frac{\sigma_e}{\omega\varepsilon_0}$$

where $\varepsilon_{relax}$ represents the loss associated with molecular relaxation , $\sigma_{dc}$ is the electrical conductivity at zero frequency, $\omega$ the angular frequency, $\varepsilon_0$ the permittivity of the free space, and $\sigma_e = \kappa'' \cdot \omega \varepsilon_0$ is the effective conductivity, which represent the TDR-measured electrical conductivity of the material (Topp et al. 2000). Therefore, measuring $\kappa_a$ TDR allows to get simultaneously the dielectric constant and the effective electrical conductivity. There is plenty of literature showing how this is possible (Dalton et al. 1984; Topp et al. 1988; Weerts et al. 2001; Noborio 2001; Jones et al. 2002; Robinson et al. 2003; Lin et al. 2007; Thomsen et al. 2007; Huisman et al. 2008; Lin et al. 2008; Koestel et al. 2008; Bechtold et al. 2010; and many others). In summary, neither EMI nor TDR directly measure the electrical conductivity but both of them allow to retrieve the electrical conductivity by the apparent conductivity using inversion and/or calibration.

When the reviewer says that the use of TDR to measure the electrical conductivity is not a recommended approach he probably means that TDR doesn't allow sufficient accuracy and/or he questions how the dielectric constant and the electrical conductivity are correlated, being estimated from one measurement (the apparent permittivity). In such cases, we understand the reviewer's concerns. However, to this point too there is a lot of literature that partially disagrees with him. Huisman et al. (2008), Lin et al. (2008), Koestel et al. (2008), and Bechtold et al. (2010) are just some examples. When the bulk electrical conductivity is in the range 0.02 – 2 dS/m (which is the range of bulk electrical conductivities we measured in our case) and when TDR is properly used (for example, good installation of the probes minimizing the effect of nonparallel wires; minimization of the soil disturbance), it allows electrical conductivity measurements with errors less than 5% (Huisman et al. 2008; Bechtold et al. 2010). In such cases, TDR electrical conductivity measurements reach accuracies so high to make TDR a potential tool to rate the validity of the electrical conductivity data obtainable from EMI data inversion. Moreover, with such high accuracy TDR can act as a quantitative ground truth for ERT too, as Koestel et al. state in their conclusion: "In addition, the results suggest that TDR has high potential to act as a quantitative ground truth for ERT. Furthermore, it may be possible to use TDR data to constrain the ERT inversion process."

Given the above, we feel that the use of the TDR to calibrate EMI data is totally defendable and it is actually a technique that can provide ground truth as reliable as ERT for electrical conductivity.

Anyway, since the intent of our paper is neither setting up TDR against ERT nor proving whether TDR is more reliable than ERT or not, we will revise the manuscript with the only aim of presenting the TDR as a viable and reliable calibration tool for EMI data.

2. The explanation of the filtering procedure we used to make TDR and EMI measurements comparable.

*Spending time describing Fourier transformation is probably useless. Rather, I would concentrate on describing in detail what type of filtering is applied. "Fourier filtering" is unclear. I presume it is a spatial filtering made to enhance the long wavelengths? Please be more specific and try and link the approach to established (there are far too many) filtering techniques.*

Totally agreeing with the reviewer about this point, we will update the manuscript, removing all unnecessary details from the long description of the Fourier transformation and describing the filtering procedure in more detail. Since the filtering is a key point in our calibration procedure,

we anticipate here a brief schematic description to explain how we filtered the TDR series to make them comparable with the EMI series.

1) For each of the four transects, we have preliminary retrieved by EMI (sharp) inversion three series (horizontal profiles) of EMI bulk conductivity ($\sigma_{,EMI}$), one for each depth interval for which TDR series are available (0-20 cm, 20-40 cm, and 40-60 cm). To this end, for each EMI sounding we have averaged the conductivities in each of the depth intervals. Then, we have estimated the mean and the standard deviation for the EMI conductivity along these horizontal profiles.

2) Firstly, a zero-padded version of the measured TDR data have been converted from the spatial domain to the wavenumber domain (Fourier spectrum) using the Fast Fourier Transform (FFT). Then, these spectra have been multiplied by the wavenumber response function of low-pass filters. Finally, the spectrum of each output has been converted back from the wavenumber domain to the spatial domain using the Inverse FFT to get the filtered TDR data.

3) Original TDR profiles have been repeatedly filtered using low-pass filters with different low-cut wavenumbers, from the Nyquist to 0 cycles/m.

4) By comparing the standard deviation of the filtered TDR profiles to the standard deviation of the corresponding EMI profiles, we were able to select the optimum filtered TDR profile to calibrate EMI data.

Just as an example, Figure 1 shows the result of the described procedure for the 50-6dS transect at 20-40 cm. In this case, the cut-off frequency to make the two standard deviations similar corresponds to 0.313 cycles/m. In other words, this means that the two series are characterized by similar patterns of variability at 3.2 m distance.

[Figure]

Figure 1

**Other remarks**
*Line 26: "contributing to enhance the spatial resolution of the EMI reconstruction". I am not sure one can claim that the use of a stabilizer (how much needed would also require a specific discussion) truly*

*enhances spatial resolution of a geophysical method. In my opinion this statement is wrong. I suggest a reformulation here.*

The statements about the "spatial resolution" will be reformulated in the revised manuscript.

*Line 125: "Then we assess the quality of these reconstructions by using TDR data as ground-truth." This is a very brave statement. I do not see TDR as any more reliable to measure sigma than EMI, indeed quite the opposite.*
*Line 132: "Accordingly, the paper provides a methodology to calibrate EMI results by TDR readings." This should not (cannot) be the focus of this paper. If the authors believe this is a viable strategy, I totally disagree.*
*Figure 6: the difference between TDR and EMI measured sigma is quite large indeed. Overall I am not sure that TDR is the best method to measure sigma. Indeed it is not. TDR is the chief approach to measure dielectric properties.*

We fully discussed these issues above.

*Line 727 Figure 2. "Examples of sharp and smooth inversions applied to the same dataset 100-6dS. The results are shown together with their corresponding data misfit". I see only one curve of data misfit. Does it refer to both sharp and smooth inversion?*

Indeed, Figure 2 shows both sharp and smooth inversion data misfit curves, although they are barely discernible due to the poor quality of the figure. In the revised paper the quality of the figure will be improved in order to make curves clearly discernible.

*Figure 8: the difference between the two images is striking. I am not sure how the authors are so confident that the correction applied to obtain the revised EMI image is correct.*

Figure 8 was wrong. We will update it in the revised manuscript.

**References**

Bechtold, M., Huisman, J.A., Weihermüller, L., Vereecken, H. (2010). Accurate determination of the bulk electrical conductivity with the TDR100 cable tester. Soil Sci. Soc. Am. J. 74, 495–501

Dalton, F. N., Herkelrath, W. N., Rawlins, D. S., and Rhoades, J. D. (1984). Time domain reflectometry: Simultaneous measurement of soil water content and electrical conductivity with a single probe. Science 224, 989–990.

Huisman, J.A., Lin, C.P., Weihermüller, L., Vereecken, H. (2008). Accuracy of Bulk Electrical Conductivity Measurements with Time Domain Reflectometry. Vadose Zone J. 7, 426–433

Jones, S.B., Wraith, J.M., Or, D. (2002). Time Domain Reflectometry (TDR) measurement principles and applications. Hydrol. Proc. 16:141-153.

Koestel, J., Kemna, A., Javaux, M., Binley, A., Vereecken, H. (2008). Quantitative imaging of solute transport in an unsaturated and undisturbed soil monolith with 3-D ERT and TDR, Water Resour. Res., 44, W12411, doi:10.1029/2007WR006755.

Lin, C.P., Chung, C.C., Huisman, J.A., Tang, S.H. (2008). Clarification and calibration of reflection coefficient for time domain reflectometry electrical conductivity measurement. Soil Sci. Soc. Am. J. 72, 1033-104072.

Lin, C.P., Chung, C.C., Tang, S.H. (2007). Accurate time domain reflectometry measurement of electrical conductivity accounting for cable resistance and recording time. Soil Sci. Soc. Am. J. 71:1278–1287.

McNeill, J.D. (1980). Electromagnetic terrain conductivity measurement at low induction numbers. Technical Note TN-6. Geonics Ltd, Mississauga, ON, Canada

Nadler A., Dasberg, S., Lapid, I. (1991). Time Domain Reflectometry Measurements of Water Content and Electrical Conductivity of Layered Soil Columns. Soil Sci. Soc. Am. J. 55:938-943

Noborio, K. (2001). Measurement of soil water content and electrical conductivity by time domain reflectometry: A review. Comput. Electron. Agric. 31:213–237.

Ren T., Noborio, K., Horton, R. (1999). Measuring Soil Water Content, Electrical Conductivity, and Thermal Properties with a Thermo-Time Domain Reflectometry Probe. Soil Sci. Soc. Am. J. 63:450–457

Robinson, D. A., Jones, S. B., Wraith, J. M., Or, D., Friedman, S.P. (2003). A review of advances in dielectric and electrical conductivity measurement using time domain reflectometry. Vadose Zone J. 2, 444–475.

Thomsen, A., Schelde, K., Drøscher, P., Steffensen, F. (2007). Mobile TDR for geo-referenced measurement of soil water content and electrical conductivity. Precision Agriculture 8, 213–223

Topp, G. C., Yanuka, M., Zebchuk, W. D., and Zegelin, S. (1988). Determination of electrical conductivity using time domain Reflectometry: Soil and water experiments in coaxial lines. Water Resour. Res. 24, 945–952.

Topp, G.C., Zegelin, S., White, I., (2000). Impacts of the real and the imaginary components of relative dielectric permittivity on time domain reflectometry measurements in soils. Soil Sci. Soc. Am. J. 64, 1244–1252.

Weerts A. H., Huisman, J.A., Bouten, W. (2001). Information content of time domain reflectometry waveforms. Water Resources Research, vol. 37, n. 5, 1291–1299

---

## Author Comment (AC2) · 15 Sep 2017

**Reply to Reviewer #2**

The response to the individual comments of Reviewer #2 is given below. In the following, the original review is quoted in *italics*.

*TDR conductivity measurements show larger conductivity close to the surface as it should be expected since it is closer to the source of conductive material (added saline water). For this aspect, it is acceptable to consider TDR conductivity measurements as providing more reliable information, which could serve to calibrate "less reliable conductivities" obtained with EMI measurements (or their inversion). I therefore agree with the overall approach. I find the paper clear and well-written. I have nevertheless few questions at the EMI inversion stage which, I think, should be explored, or at least discussed before acceptance. I also think that Authors should show the final results for the other transects in order to see if the calibration performs well for the different irrigation experiments. This being said, this manuscript is interesting as it addresses the problem of relating ground truth and EMI output with a pragmatic approach.*

We agree with the Reviewer's remarks: in the revised version of the manuscript we will further elaborate on the role and importance of the assumptions made during the inversion stage and we will also show the results for the others transects.

*Non-uniqueness of the conductivity model resulting from the inversion:*
*One particular choice of the present study (compared to other cited studies) is to calibrate the conductivity after inversion instead of EMI apparent conductivity data (Eca). However, the argument of non-uniqueness of the inverse problem, which is actually used by the authors in the introduction to question a calibration with ERT method, could be used here in the same way to question the presented method.*

During the preparation of the new version, we will stress this aspect and we will make our point clearer: all the limitations connected with the ill-posedness are inherited by our approach that, indeed, is dealing merely with the absolute calibration of the model derived by the EMI data, based on the TDR measurements. For this reason, and accordingly to the Reviewer's remark, we will further elaborate on the fact that the output of our approach is still relying on the hypothesis made during the EMI data inversion.

*This said, are you sure that the selected solution obtained using sharp regularization is the best solution to be compared with TDR? Actually, some of the smooth models (e.g. sounding numbers 5, 13 17 in Figure 2) show better vertical concordance with what should be expected and with the best misfit. Anyway, Figure 2 clearly highlights the non-uniqueness of the problem, because it is possible to obtain very different models at similar misfits. Why, for example, not putting some effort to stabilize the regularization of the smooth inversion in order to have sounding N5-like results all along the transect? I do see that, because of fixed interfaces and a very little number of layers, your parameterization does not allow enough model space flexibility to have stable smooth results along the transect. But there are many ways to fix such issues (like, for example, among others, increasing the number of layers or applying some lateral constraints). I would also like to point out that the smooth misfits are slightly higher than for the sharp method (of 1-2 %, except for the few soundings mentioned above. This feature supports my previous comments), which is not in favor of the smooth regularization in a context of fair comparison. For all this, I would not qualify the smooth inversions presented here as a "standard" approach as it is not used with an optimal way (you use few layers with fixed interfaces). For Figure 2, I*

*really recommend to plot the profiles of collected EMI data together with modeled (after inversion) data to see exactly how well all the channels were fitted. This would allow to properly evaluate and discuss the two methods of inversions. This is also important for a second aspect: in a cultivated area, I would expect some lateral anisotropy of the conductivity of the soil layer due the preferential orientation of the lines of the agricultural work. If this is true, inverting HCP and VCP together would not be an optimal choice as such geometries induce eddy current with different preferential directions. Did you try to consider HCP only, and/or VCP only? Maybe there would be a better qualitative and quantitative consistency between TDR and EMI inversion results?*

The Reviewer correctly highlighted that we used fix interfaces in our model parameterization. However, we have used a discretization with several tens of layers to be able to: (i) control the inversion results by acting only on the regularization parameters and (ii) remove the regularization effects possibly originated by the discretization choices (e.g., the number of layers, interfaces locations). In this way, we have been able to use an automatic strategy for the selection of the regularization parameters. For these reasons, we believe that our smooth result can be defined "standard".

Regarding the use of lateral constraints, it can definitely be a viable solution, but we showed that a satisfactory lateral consistency can be achieved by using an already existing 1D code (slightly modified to accommodate a sharp regularization) instead of implementing a (clearly more troublesome to code) pseudo-2D (e.g., laterally constrained) version of it.

Moreover, from our point of view, the fact that the data misfits are largely overlapping, confirms that the two inversion results are actually comparable. And, if the sharp inversion fits better the data, a simple explanation might be that the assumption of sharp interfaces is in a better agreement with the reality and that the (blocky) true model is difficult to be correctly retrieved when smooth constraints are applied.

We agree with the reviewer that the best way to assess the quality of the data fitting is to plot the collected data against the calculated ones. So, in the revised manuscript we will add a plot and the associated discussion about it.

*To sum up, I would have interpreted the results shown in Figure 2 in a different manner.*
*After acceptation of the non-uniqueness of the EMI data inversion, I would have tried to find the right parameterization and regularization to get sounding N5-like results along the transect before starting the comparison with TDR. As a consequence, I believe that the calibration procedure presented in this study also corrects error due to the non-uniqueness inherent to the considered inverse problem (in addition to the spatial fractality problem already discussed in the text). And in the eventual case of lateral anisotropy, it maybe also correct for less realistic EMI results resulting from joint HCP and VCP inversion. However this two features means that the overall calibration procedure could be dependent on the initial method of inversion. In my opinion, this aspect should be explored in this study, or at least discussed in the text.*

As it was mentioned above, we agree with the Reviewer and we will modify the paper taking into account his/her suggestions.

*Application of the method to the four transects:*
*I think you should show the resulting sections (like Figure 8) for all the transects in order to check the consistency of the calibration procedure over larger areas (and implicitly for broader geological/irrigation settings) as it is claimed in the conclusion.*

Also regarding this specific point, we see the rationale behind Reviewer's remark and, in the new version, we will show the results from other transects.

*Some minor comments:*
*In the discussion about magnetic permeability (p7 line 239-241), you cite a nonpublished paper (Deidda et al, submitted). Here, it is necessary to also cite already published papers on this topic. There are a couple of recent studies dealing with the inversion of in-phase data for retrieving the magnetic permeability for the case of small EMI sensors.*

In the revised version, we will include the references to additional relevant studies.

*Figure 8 shows spectacularly that the TDR conductivity of the first layer is largely underestimated by the EMI sharp inversion results. Are we sure that this first layer is well constrained by the considered EMI vertical soundings? Maybe it would be good to show and analyse the a priori covariance on model parameter associated to the selected 4-heights/2-geometries data set.*

In the new version of the manuscript, Figure 8 will be largely revised also following the observations of Reviewer #1. Accordingly, a new associated discussion will be added.

*Summary of my recommendations:*
*Plot the measured data versus the modeled data in Figure 2. Explore or discuss the dependence of the overall procedure on the method of inversion. Show the final results for the four transects to confirm the robustness of the method on different irrigation contexts.*
*Sincerely*

We thank the Reviewer for his/her useful and pertinent comments and suggestions. As mentioned above, in the new manuscript, we will show the measured vs calculated data, make our point clearer regarding the dependence on the adopted inversion strategy, present the results from other transects.

---

## Author Response (AR1)

**Response to Reviewer #1**

We are thankful to prof. Cassiani for his valuable comments and suggestions, which have certainly improved the manuscript. The response to the individual comments is given below. The original review is quoted in *italics*, whereas our response is given in **bold**.

(**1**) *I have found the paper interesting and potentially worth publication. However I find it somewhat surprising that the authors seem to believe that TDR is a better method than EMI to measure electrical conductivity. This seems to be an assumption made a priori, and not supported either by the scientific literature nor by any evidence in the paper. EMI is designed specifically to measure electrical conductivity, while TDR is designed with the measurement of dielectric properties*

*in mind. Using TDR also to measure electrical conductivity can be done, similarly to using attenuation in GPR measurements to do the same. However it is not a recommended approach. My suggestion to the author is to reverse the line of reasoning, believe more in EMI (with some caveats especially concerning the depth of investigation) and rather question TDR as a method for sigma measurement. In a nutshell, give more credibility to geophysics and question some belief in soil science. To this end, I also suggest that an eye is given to ERT as a technique that can provide ground truth much more*

*reliable that TRD for electrical conductivity (see e.g. Cassiani et al., 2012 and Ursino et al., 2014, but many other papers deal with the EMI-ERT obvious relationship). I am also very surprised that moisture content estimates from TDR are not considered at all in the paper – yet the data must be available. I suggest the authors present also those (much more solid, I presume) data. I encourage the authors to revise the paper along these lines and resubmit this potentially interesting dataset.*

**Concerning the effectiveness and accuracy of electrical conductivity measurements via TDR, consistently with our previous reply to the Reviewer, we largely extended the discussion in the revised version of the manuscript and added several references supporting the rationale behind our approach.**
**We believe that, in the new manuscript, the limits of validity of the presented workflow for the calibration of the EMI inversion results by using TDR measurements are now better defined and the associated assumptions are more**

**clearly explained.**

(**2**) *Line 26: "contributing to enhance the spatial resolution of the EMI reconstruction". I am not sure one can claim that the use of a stabilizer (how much needed would also require a specific discussion) truly enhances spatial resolution of a geophysical method. In my opinion this statement is wrong. I suggest a reformulation here.*

**With that statement, we simply mean that the appropriate regularization can increase the capability to distinguish blocky anomalies. In fact, "standard" (smoothing) stabilizers, when blindly applied to targets with sharp boundaries, generally result in a significant reduction of the resolution capabilities as they tend to smear the anomalies out. The**

contrary is true for sharp inversion algorithms (clearly only when they are applied to soils characterized by abrupt changes in the property under investigation).

However, we see the point of the Reviewer and, in the new version of the paper, we modified all the parts dealing with the enhanced resolution capabilities of the sharp inversion to better explain what we actually mean and to limit the risks of misunderstandings.

**(3)** *Line 35: "after filtering the TDR data" Even though this is the abstract, the statement is far too generic. Details about the filtering approach shall be briefly given here.*

Accordingly to the Reviewer's suggestion, in the new version of the abstract, we added more details about the TDR data filtering. Moreover, in the rest of the article, we significantly extended the part concerning the strategy to design the optimal filtering and, on the other hand, removed the long portion devoted to the description of the Fourier Transform.

**(4)** *Line 125: "Then we assess the quality of these reconstructions by using TDR data as ground-truth." This is a very brave statement. I do not see TDR as any more reliable to measure sigma than EMI, indeed quite the opposite. Line 132:"Accordingly, the paper provides a methodology to calibrate EMI results by TDR readings." This should not (cannot) be the focus of this paper. If the authors believe this is a viable strategy, I totally disagree.*

On this respect, kindly, see our previous reply to comment # **1**.

**(5)** *Line 291 and following. Spending time describing Fourier transformation is probably useless. Rather, I would concetrate on describing in detail what type of filtering is applied. "Fourier filtering" is unclear. I presume it is a spatial filtering made to enhance the long wavelengths? Please be more specific and try and link the approach to established (there are far too many) filtering techniques.*

We definitely see the Reviewer's point. As already mentioned in our reply to comment # **3**, in the new version, we removed the Fourier Transform description and focused on the discussion of the utilzed filter.

**(6)** *Line 573: "Ferre" is actually "Ferré".*

This typo is now corrected.

(**7**) *Line 727 Figure 2. "Examples of sharp and smooth inversions applied to the same dataset 100-6dS. The results are shown together with their corresponding data misfit". I see only one curve of data misfit. Does it refer to both sharp and smooth inversion? Also, I find it a bit difficult to justify in the images how some dark blue areas in the smooth inversion indeed correspond to slightly less dark blue areas in the sharp inversion. I am also a bit skeptical of the fact that using an EM38 one can image with confidence to a depth as large as 3 m!.*

**Actually, the original Fig. 2 was already showing both (sharp and smooth) data misfits for the 100-6dS case. However, in the new version, also following the suggestions of the Reviewer #2, we decided to: (i) improve the clarity of the data misfit plot of the original Fig. 2; (ii) include a similar figure for the 50-6dS case (the new Fig. 3); (iii) add a figure, for each of the 100-6dS and 50-6dS cases, showing the observed and calculated data corresponding to the sharp and smooth reconstructions. These additional figures further demonstrate the ill-posedness of the inversion problem (in particular in presence of a very limited number of observations with relatively high levels of noise). In fact, it is clear from the new Fig.s 2-5 that, despite the data fittings are very similar, the sharp and smooth inversions provide significantly different results in all the considered cases.**

**Concerning the section depth, we are not necessarily claiming that the EM38 is actually investigating down to 3 m. After all, we use only the first 0.6 m along the paper. To some extent, we decided to "over-parametrize" the model (for example, in terms of number and density of the layers) to prevent any possible side-effects and to let the regularization be uniquely controlled by the stabilizer. In the new manuscript, we have added a few lines about these aspects.**

(**8**) *Line 735, Figure 3:here too some details about the filter applied to the TDR data shall be given. It is not acceptable that in a caption only the term "filtered" is applied. One can use any type of filter! The same applies to Figures 4 and 5.*

**Please, see our reply to the comment # 3.**

(**9**) *Figure 6: the difference between TDR and EMI measured sigma is quite large indeed. Overall I am not sure that TDR is the best method to measure sigma. Indeed it is not. TDR is the chief approach to measure dielectric properties.*

**The difference in the original Fig. 6 is not surprising as, for example, it is well-known that EMI measurements require frequent calibrations (just as an example, see Lavoué et al., 2010, Electromagnetic induction calibration using apparent electrical conductivity modelling based on electrical resistivity tomography. Near Surface Geophysics 8(6),**

**553-561). And this is why we developed a calibration strategy for the EMI results based on the TDR measurements. Regarding the appropriateness of using TDR as reliable methodology to retrieve the true soil conductivity, please, see our reply to comment # 1.**

**(10)** *Figure 8: the difference between the two images is striking. I am not sure how the authors are so confident that the correction applied to obtain the revised EMI image is correct.*

**In the revised version of the manuscript, we completely modify the original Fig. 8 (corresponding now to Fig. 10). The new Fig. 10 shows the calibration results for all transects and demonstrates that the proposed strategy preserves the spatial variability of the EMI reconstruction but with conductivity ranges compatible with the TDR measurements (here assumed to be reliable estimation of the true soil conductivity).**

**Response to Reviewer #2**

We thank Reviewer #2 for his/her valuable comments and suggestions, which have certainly improved the manuscript. The response to the individual comments is given below. The original review is quoted in *italics*, whereas our response is given in **bold**.

(**1**) *TDR conductivity measurements show larger conductivity close to the surface as it should be expected since it is closer to the source of conductive material (added saline water). For this aspect, it is acceptable to consider TDR conductivity measurements as providing more reliable information, which could serve to calibrate "less reliable conductivities" obtained with EMI measurements (or their inversion). I therefore agree with the overall approach. I find the paper clear and well-*

*written. I have nevertheless few questions at the EMI inversion stage which, I think, should be explored, or at least discussed before acceptance. I also think that Authors should show the final results for the other transects in order to see if the calibration performs well for the different irrigation experiments. This being said, this manuscript is interesting as it addresses the problem of relating ground truth and EMI output with a pragmatic approach.*

**We agree with the Reviewer's remarks. Accordingly, in the revised version of the manuscript, we further elaborated on the role and importance of the assumptions made during the inversion stage and also shown the results from all the other transects.**

(**2**) *Non-uniqueness of the conductivity model resulting from the inversion:*

*One particular choice of the present study (compared to other cited studies) is to calibrate the conductivity after inversion instead of EMI apparent conductivity data (Eca). However, the argument of non-uniqueness of the inverse problem, which is actually used by the authors in the introduction to question a calibration with ERT method, could be used here in the same way to question the presented method.*

**In the new version of the paper, we stressed this aspect and made our point clearer. So, we further elaborated on the fact that all the limitations/assumptions connected with the ill-posedness of the EMI inversion are inevitably inherited by our approach.**

**We also added a brief discussion on a possible alternative strategy to use the TDR measurements to calibrate directly the EMI-EC$_a$ data (data-space calibration). This second option would not need any statistical analysis of the data**

**since the physics of the EMI forward modelling would effectively perform the rescaling. However, eventually, also in this case, when it is time to translate the (now calibrated) EMI-EC$_a$ into the corresponding conductivities $\sigma_{b,EMI}$, it will be necessary to go through the inversion process (with all its associated assumptions).**

**As it is discussed in the new manuscript, we opted for the model-space calibration for pragmatic reasons connected with the possible difficulties to decuple the forward modelling parts in the, usually available, EMI inversion codes.**

**(3)** *This said, are you sure that the selected solution obtained using sharp regularization is the best solution to be compared with TDR? Actually, some of the smooth models (e.g. sounding numbers 5 , 13 17 in Figure 2) show better vertical concordance with what should be expected and with the best misfit. Anyway, Figure 2 clearly highlights the non-uniqueness of the problem, because it is possible to obtain very different models at similar misfits. Why, for example, not putting some effort to stabilize the regularization of the smooth inversion in order to have sounding N5-like results all along the transect? I do see that, because of fixed interfaces and a very little number of layers, your parameterization does not allow enough model space flexibility to have stable smooth results along the transect. But there are many ways to fix such issues (like, for example, among others, increasing the number of layers or applying some lateral constraints). I would also like to point out that the smooth misfits are slightly higher than for the sharp method (of 1-2 %, except for the few soundings mentioned above. This feature supports my previous comments), which is not in favor of the smooth regularization in a context of fair comparison. For all this, I would not qualify the smooth inversions presented here as a "standard" approach as it is not used with an optimal way (you use few layers with fixed interfaces). For Figure 2, I really recommend to plot the profiles of collected EMI data together with modeled (after inversion) data to see exactly how well all the channels were fitted. This would allow to properly evaluate and discuss the two methods of inversions. This is also important for a second aspect : in a cultivated area, I would expect some lateral anisotropy of the conductivity of the soil layer due the preferential orientation of the lines of the agricultural work. If this is true, inverting HCP and VCP together would not be an optimal choice as such geometries induce eddy current with different preferential directions. Did you try to consider HCP only, and/or VCP only? Maybe there would be a better qualitative and quantitative consistency between TDR and EMI inversion results?*

**The Reviewer correctly highlighted that we used fix interfaces in our model parameterization. However, we have used a discretization with 100 layers down to 8 m depth to be sure: (i) to control the inversion results by acting only on the regularization parameters and (ii) to remove the regularization effects possibly originated by the discretization choices (e.g., the number of layers, interfaces locations). In this way, we have been able to use an automatic strategy for the selection of the regularization parameters. For these reasons, we believe that our smooth result can be defined "standard". We added to the new manuscripts a few lines about this to better explain our point.**

**Regarding the use of lateral constraints, it can definitely be a viable solution, but we showed that a satisfactory lateral consistency can be achieved by using an already existing 1D code (slightly modified to accommodate a sharp regularization) instead of implementing a (clearly more troublesome to code) pseudo-2D (e.g., laterally constrained) version of it.**

**Moreover, from our point of view, the fact that the data misfits are largely overlapping, confirms that the two inversion results are actually comparable. And, if the sharp inversion fits better the data, the simplest explanation**

**might be that the assumption of sharp interfaces is in a better agreement with the reality and that the (blocky) true model is difficult to be correctly retrieved when smooth constraints are applied.**

**These arguments are confirmed by the results for all transects. On this respect, in the revised paper, we included an additional new figure (Fig. 4) showing a consistent behaviour also for the 50-6dS case.**

**We totally agree with the reviewer that the best way to assess the quality of the data fitting is to plot the collected data against the calculated ones. So, in the revised manuscript, we added two new figures on this respect (Fig.s 3 and 5).**

**(4)** *To sum up, I would have interpreted the results shown in Figure 2 in a different manner. After acceptation of the non-uniqueness of the EMI data inversion, I would have tried to find the right parameterization and regularization to get*
*sounding N5-like results along the transect before starting the comparison with TDR. As a consequence, I believe that the calibration procedure presented in this study also corrects error due to the non-uniqueness inherent to the considered inverse problem (in addition to the spatial fractality problem already discussed in the text). And in the eventual case of lateral anisotropy, it maybe also correct for less realistic EMI results resulting from joint HCP and VCP inversion. However this two features means that the overall calibration procedure could be dependent on the initial method of inversion. In my*
*opinion, this aspect should be explored in this study, or at least discussed in the text.*

**Kindly see our previous replies to comments # 1-3.**

**(5)** *Application of the method to the four transects:*
*I think you should show the resulting sections (like Figure 8) for all the transects in order to check the consistency of the calibration procedure over larger areas (and implicitly for broader geological/irrigation settings) as it is claimed in the conclusion.*

**We definitely see the rationale behind Reviewer's remark and, in the new version, we showed the results from all**
**transects. They confirm the robustness and reliability of the proposed calibration approach.**

**(6)** *Some minor comments:*
*In the discussion about magnetic permeability (p7 line 239-241), you cite a nonpublished paper (Deidda et al, submitted). Here, it is necessary to also cite already published papers on this topic. There are a couple of recent studies dealing with the*
*inversion of in-phase data for retrieving the magnetic permeability for the case of small EMI sensors.*

**Following the Reviewer's suggestion, we included the references to additional relevant studies.**

(**7**) *Figure 8 shows spectacularly that the TDR conductivity of the first layer is largely underestimated by the EMI sharp inversion results. Are we sure that this first layer is well constrained by the considered EMI vertical soundings? Maybe it would be good to show and analyze the a priori covariance on model parameter associated to the selected 4-heights/2-geometries data set.*

**The original Fig. 8 (now Fig. 10) has been largely revised, and, now, it is not subdivided into three layers anymore. The new final results (Fig. 10d-g) show significant improvements as they effectively merge the information contents of the two original datasets. In fact, at the same time, they preserve the spatial variability (over relatively large areas) of the EMI model together with the reliable conductivity ranges supplied by the TDR measurements.**

(**8**) *Summary of my recommendations:*
*Plot the measured data versus the modeled data in Figure 2. Explore or discuss the dependence of the overall procedure on the method of inversion. Show the final results for the four transects to confirm the robustness of the method on different irrigation contexts.*

**Once again, we thank the Reviewer for his/her useful and pertinent comments and suggestions. As mentioned above, in the new manuscript, we showed the measured vs calculated data, made our point clearer regarding the dependence on the adopted inversion strategy, presented the results from the other transects.**

[revised manuscript text omitted]

---

## Author Response (AR2)

**Response to the Editor**

We are thankful to prof. Zehe for his supervision of the review process, which have certainly improved the manuscript. The response to the individual comments is given below. The original remarks are quoted in *italics*, whereas our response is given in **bold**.

5

(1) I am delighted to let you know that both reviewers are pleased with the revised version of your manuscript. Reviewer 2 recommends a few technical corrections, which I believe you are willing to implement. In line with her/him I suggest that you might consult the references listed below and optionally refer to those in case you find them or relevance.

10 Sasaki, Y., J. Kim, and S. Cho, 2010, Multidimensional inversion of loop-loop frequency domain EM data for resistivity and magnetic susceptibility: Geophysics, 75, 213-223.

Guillemoteau, J., F. X. Simon, E. Luck, and J. Tronicke, 2016, 1D sequential inversion of portable multi-configuration electromagnetic induction data: Near Surface Geophysics, 14, 411-420.

15

Noh, K., K. H. Lee, S. Oh, S. J. Seol, and J. Byun, 2017, Numerical evaluation of active source magnetics as a method for imaging high-resolution near-surface magnetic heterogeneity: Geophysics, 82, J27-J38.

As suggested, in the new version of the manuscript we included the additional references.

20 Moreover, in the revised manuscript, we carefully considered the recommendations provided by Reviewer #2 and provided a point-by-point reply to her/his suggestions.

**Response to Reviewer #2**

10

We thank Reviewer #2 for his/her valuable comments and suggestions. The response to the individual comments is given below. The original review is quoted in *italics*, whereas our response is given in **bold**.

**5 (1) I have few additional minor-moderate recommendations.**

The authors added the data plots so that the readers can now properly evaluate the quality of the geophysical data and their inversion.

In Figures 2 and 4, I nevertheless recommend decreasing the upper limit of the color-bar to 0.05-0.06 S/m. For now, the color-scale gives the impression that the smooth solution is rather homogeneous, which, sometimes, is not true: it is just varies within the blue color range.

Figure R2: Examples of sharp and smooth inversions applied to the dataset 100-6dS. The results are shown together with their corresponding data misfit.